# OPENDAS: OPEN-VOCABULARY DOMAIN ADAPTATION FOR 2D AND 3D SEGMENTATION

## ABSTRACT

Recently, Vision-Language Models (VLMs) have advanced segmentation techniques by shifting from the traditional segmentation of a *closed-set* of predefined object classes to *open-vocabulary* segmentation (OVS), allowing users to segment *novel* classes and concepts unseen during training of the segmentation model. However, this flexibility comes with a trade-off: fully-supervised closed-set methods still outperform OVS methods on *base* classes, that is on classes on which they have been explicitly trained. This is due to the lack of pixel-aligned training masks for VLMs (which are trained on image-caption pairs), and the absence of domain-specific knowledge, such as autonomous driving. Therefore, we propose the task of *open-vocabulary domain adaptation* to infuse domain-specific knowledge into VLMs while preserving their open-vocabulary nature. By doing so, we achieve improved performance in base *and* novel classes. Existing VLM adaptation methods improve performance on base (training) queries, but fail to fully preserve the open-set capabilities of VLMs on novel queries. To address this shortcoming, we combine parameter-efficient prompt tuning with a triplet-loss-based training strategy that uses auxiliary negative queries. Notably, our approach is the only parameter-efficient method that consistently surpasses the original VLM on novel classes. Our adapted VLMs can seamlessly be integrated into existing OVS pipelines, e.g., improving OVSeg by +6.0% mIoU on ADE20K for open-vocabulary 2D segmentation, and OpenMask3D by +4.1% AP on ScanNet++ Offices for open-vocabulary 3D instance segmentation without other changes.

## 1 INTRODUCTION

Recent developments in Vision-Language Models (VLMs), such as CLIP (Radford et al., 2021) or SigLIP (Zhai et al., 2023), catalyzed a paradigm shift in visual understanding. They have enabled significant advances in detection, localization, and segmentation from open-vocabulary queries.

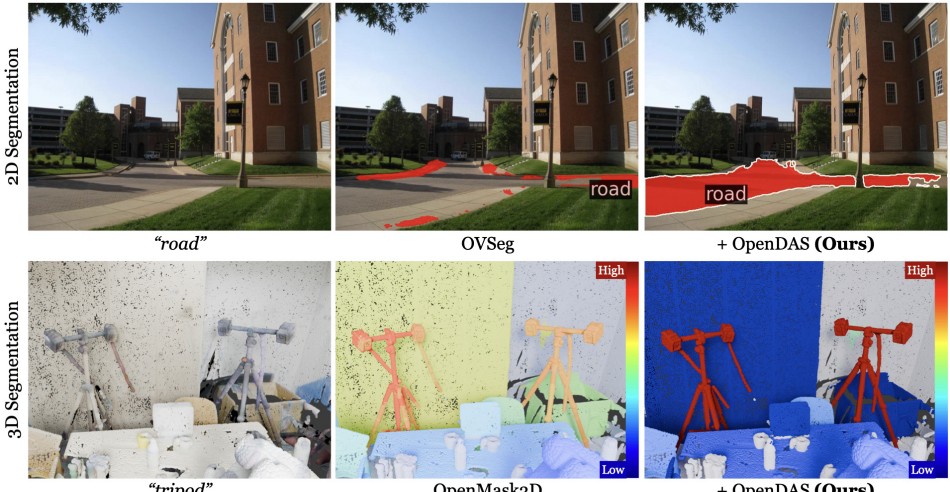

**Figure 1: Open-Vocabulary Domain Adaptation for Segmentation.** We adapt VLMs to new domains while preserving their open-vocabulary nature, and integrate them to existing OVS pipelines such as OVSeg (Liang et al., 2023) *(top)* and OpenMask3D (Takmaz et al., 2023a) *(bottom)*. We show the results with a seen query "road" for 2D and the similarity score with an unseen query "tripod" for 3D, with red indicating high similarity.

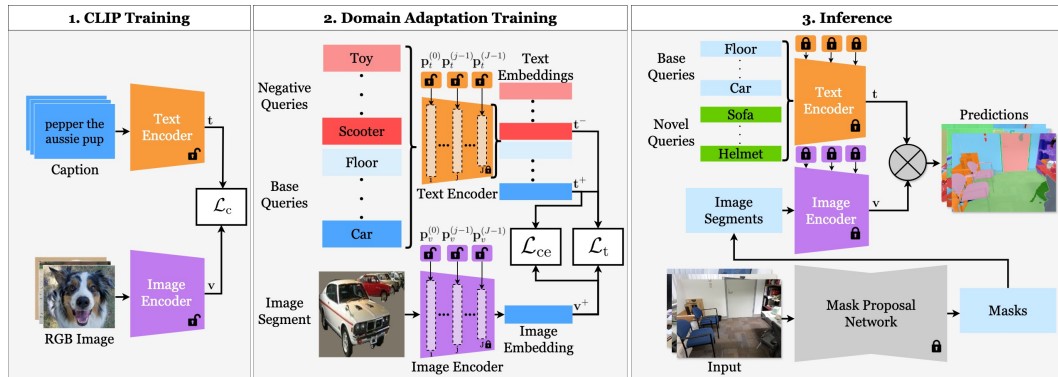

**Figure 2: Illustration of the OpenDAS architecture.** *Left:* Our work builds on CLIP (Radford et al., 2021), a VLM pre-trained on image-caption pairs with a contrastive loss $\mathcal{L}_c$. *Center:* We adapt the CLIP text- and image-encoders using prompt tuning with base (training) queries and generated negative queries to inject domain-specific priors. We insert visual prompts, $\mathbf{p}_v^{(0)}, ..., \mathbf{p}_v^{(J-1)}$, and textual prompts, $\mathbf{p}_t^{(0)}, ..., \mathbf{p}_t^{(J-1)}$, to the input of the encoder layers, $1, ..., J$. We combine cross-entropy loss $\mathcal{L}_{ce}$ with triplet loss $\mathcal{L}_t$ and negative queries to enhance CLIP's performance on novel (unseen) queries. *Right:* We integrate our model to existing OVS pipelines, *i.e.*, OVSeg for 2D and OpenMask3D for 3D and test it with visually similar domains and novel queries, showing its open-vocabulary understanding capabilities while still adapting to the target domain.

Methods for open-vocabulary segmentation (OVS) leverage the open-set capabilities of VLMs, allowing them to segment novel queries not seen during training. This had a transformative impact on practical applications, ranging from household robots capable of understanding textual commands for object interaction (Lemke et al., 2024; Liu et al., 2024; Wu et al., 2023) to localization and navigation systems like Text2Loc (Xia et al., 2023) and Language Frontier Guide (Shah et al., 2023).

However, VLM-based segmentation models, which rely on text queries, underperform compared to fully supervised domain-specific models trained on fixed categories. A primary obstacle to enhancing the performance of VLMs is their reliance on extensive datasets to learn a comprehensive representation space. This presents a significant challenge as it is infeasible to manually collect millions of segments from a specialized domain while accounting for the full range of potentially novel user queries. Further, VLMs like CLIP (Radford et al., 2021) cannot effectively distinguish between items that frequently co-appear in the same image, e.g., "picture" and "frame" or "door" and "door frame". This is because CLIP is trained on image-caption pairs where a single caption describes multiple objects in the same image, leading to entangled representations that hinder precise identification and segmentation. Consequently, while CLIP exhibits robust open-set capabilities, they lack the necessary precision for specialized segmentation tasks and fine-grained distinction of objects.

To address these challenges, we introduce a new task "open-vocabulary domain adaptation for segmentation". Similar to standard domain adaptation (Farahani et al., 2020), we aim to reduce the performance gap between a source and a target domain. In our case, the source domain is the web-scale image-caption pairs used to train CLIP (Radford et al., 2021) and the target domain consists of labeled segments from three different datasets covering offices, homes, and urban streets. Unlike conventional supervised domain adaptation, our approach does not assume a closed-set vocabulary in the target domain. Specifically, the objective is to enhance language-queried object segmentation by adapting VLMs to specific target domains and annotation styles while maintaining their ability to generalize to novel language queries. This capability is crucial for practical applications, such as enabling household robots to adapt to their environments and respond to arbitrary language queries.

To solve this new task, we investigate existing OVS models for both 2D and 3D segmentation and identify the limitation of decoupled OVS methods, their reliance on VLMs' segment and text matching capabilities in the target domain. For more accurate segment classification, we explore prompt tuning methods as they are shown to be effective for task adaptation (Jia et al., 2022; Khattak et al., 2024; 2023a;b; Lee et al., 2023; Zhou et al., 2022c;d) and domain adaptation of VLMs (Gan et al., 2023; Gao et al., 2023; Jin et al., 2023). Although previous prompt tuning methods are parameter- and data-efficient, they often degrade VLM's performance on novel queries when trained with a set of base queries and image segments from a specialized domain. Hence, we propose *OpenDAS*, a novel prompt tuning method for open-vocabulary domain adaptation. Our approach (Fig. 2) uses densely labeled images with additional negative queries and a triplet loss to adapt to the target do-

main while boosting the generalization to novel queries. OpenDAS leverages state-of-the-art open-vocabulary segmentation architectures, *i.e.*, OVSeg (Liang et al., 2023) and OpenMask3D (Takmaz et al., 2023a) by replacing their CLIP-based foundation with a plug-and-play adapted VLM.

In experiments, on three challenging indoor and outdoor datasets, OpenDAS outperforms previous prompt tuning methods on both base queries and novel queries. Our proposed training strategy preserves the structure of the original CLIP embedding space by adapting the image encoder with a frozen text encoder in the first stage followed by language adaptation with a triplet loss in the second stage. Finally, we demonstrate that our approach can readily be integrated into existing OVS methods, boosting scene understanding in both 2D images and 3D scenes.

In summary, our main contributions are as follows:
- We introduce a new task, namely open-vocabulary domain adaptation for segmentation.
- We propose a simple yet effective prompt-tuning method for open-vocabulary segmentation. Combined with a novel triplet-loss-based training strategy, we further boost open-vocabulary understanding of the adapted model.
- Our method significantly outperforms existing prompt tuning methods and surpasses CLIP's understanding of novel text queries in target domains.

## 2 RELATED WORK

**2D Open-Vocabulary Segmentation.** 2D Open-Vocabulary Segmentation (OVS) consists of segmenting objects in images as specified by a user-provided language query. The common approach is to generate class-agnostic masks and visual embeddings by encoding the masks using the VLM image encoder. These are then compared to VLM text embeddings of the user query (Wu et al., 2024). For example, LSeg (Li et al., 2022) uses CLIP text embeddings and aligns pixel-level features to the text encoding of semantic class names, while OpenSeg (Ghiasi et al., 2022) aligns segment-level features with text embeddings via region-word grounding. Other approaches similarly rely on CLIP to generate text embeddings and encode images or segments in the same latent space (Cho et al., 2023; Ding et al., 2022; 2023; Liang et al., 2023; Xu et al., 2023a; Zhou et al., 2022a). These methods are inherently only as powerful as CLIP, and might, for example, fail to segment classes that often occur together in the same frame such as a door and its frame.

**3D Open-Vocabulary Segmentation.** Recent advances in 3D segmentation (Kreuzberg et al., 2022; Takmaz et al., 2023b; Weder et al., 2024; Yue et al., 2024; Huang et al., 2024), and inspired by the progress in 2D, are reshaping how we understand complex 3D scenes (Chen et al., 2024; Engelmann et al., 2024; Kerr et al., 2023; Kobayashi et al., 2022; Peng et al., 2023). As many of these methods rely on CLIP (Radford et al., 2021), its adaptation to a target domain could enhance the 3D OVS performance within the domain. Thus, we show our method's potential with an open-vocabulary 3D instance segmentation method, OpenMask3D (Takmaz et al., 2023a), which uses class-agnostic mask proposals (Schult et al., 2023) and pre-trained CLIP (Radford et al., 2021).

**Domain Adaptation and Downstream Task Adaptation.** Domain adaptation aims to align the disparity between an original training data distribution and a target domain distribution (Farahani et al., 2020). Recently, prompt tuning methods were adopted for domain adaptation to inject domain priors to the model without exhaustive full model fine-tuning (Gan et al., 2023; Gao et al., 2023; Jin et al., 2023). Moreover, prompt tuning has also been widely used for downstream task adaptation of foundation models in a parameter-efficient manner (Jia et al., 2022; Shen et al., 2024; Zhou et al., 2022c;d). Hence, we adopt prompt tuning methods to inject domain priors into VLMs and adapt them to the open-vocabulary segmentation task.

**Prompt Tuning.** Prompt tuning adds learnable parameters to the input of encoder layers to enhance model performance for specific tasks or domains. Initially proposed for LLMs (Gu et al., 2022; Lester et al., 2021; Li & Liang, 2021; Liu et al., 2023), it allows model adaptation with minimal computational costs, avoiding full model fine-tuning. Recently, it has been extended to VLMs like CLIP (Radford et al., 2021), showing promising results (Huang et al., 2022; Zhou et al., 2022d;c; Jia et al., 2022; Khattak et al., 2023a; 2024; 2023b; Lee et al., 2023). Significant contributions in unimodal prompt tuning include CoCoOp (Zhou et al., 2022c) and VPT (Jia et al., 2022). CoCoOp adds dynamic textual prompts based on image features, while VPT adds learnable visual prompts with linear probing. Recent works (Khattak et al., 2023b; Lee et al., 2023) employ multimodal learning by jointly training textual and visual prompts. MaPLe (Khattak et al., 2023a) couples

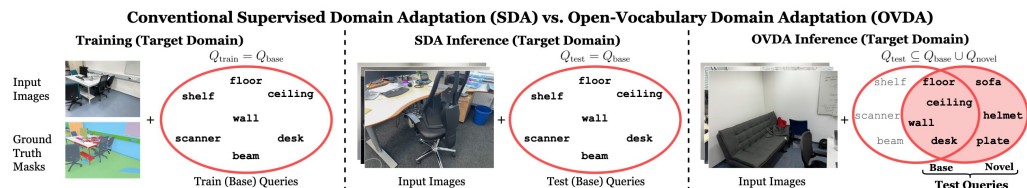

**Figure 3: Supervised Domain Adaptation (SDA) vs. our Open-Vocabulary Domain Adaptation (OVDA).** Supervised domain adaptation assumes the same vocabulary at training and test time, *i.e.*, $Q_{\text{train}} = Q_{\text{test}} = Q_{\text{base}}$. We introduce *open-vocabulary domain adaptation*, where we expect the model to learn from training (base) queries, $Q_{\text{train}} = Q_{\text{base}}$, in the target domain and respond to unseen (novel) queries, $Q_{\text{novel}}$, at test time.

interim textual and visual prompts. Instead, we use a simpler architecture with separate prompts for sequential learning providing full control over the adaptation of vision and language modalities. This significantly reduces learnable parameters while maintaining the generalization to novel queries.

## 3 OPEN-VOCABULARY DOMAIN ADAPTATION FOR SEGMENTATION

In this work, we propose the task "open-vocabulary domain adaptation for segmentation". Given a pre-trained VLM $\theta$ and a paired set of segmented images and language queries $\{I_{\text{train}}, Q_{\text{train}}\}$ for adaptation, the goal is to produce an adapted VLM $\theta^*$ that, at inference time, accurately matches image segments $I_{\text{test}}$ ($I_{\text{test}} \cap I_{\text{train}} = \emptyset$) with test-time queries $Q_{\text{test}}$. As shown in Fig. 3, traditional supervised domain adaptation methods assume access to all possible queries at training time: $Q_{\text{test}} = Q_{\text{train}} = Q_{\text{base}}$. Instead, for *open-vocabulary* domain adaptation, we define that test-time queries $Q_{\text{test}}$ can be arbitrary and are drawn both from *base* queries $Q_{\text{base}}$ (seen during adaptation training) and *novel* queries $Q_{\text{novel}}$ (not seen during adaptation training), *i.e.*, $Q_{\text{test}} \subseteq Q_{\text{base}} \cup Q_{\text{novel}}$.

The proposed task requires adapting VLMs trained on internet-scale data to a target domain with precise class labels and the task of segmentation. These domains could be indoor scenes, like offices and homes, or outdoor scenes like urban driving. As the adapted VLM can be incorporated into any mask proposal generator, our method is agnostic to the type of segmentation task.

At the same time, the model's open-vocabulary ability needs to be retained, ensuring it can accurately process language queries not seen during adaptation training. Our model is exposed to a set of training images $I_{\text{train}}$ and queries $Q_{\text{train}}$, while during inference, it can be queried with any (novel) language query $Q_{\text{base}} \cup Q_{\text{novel}}$. This requires open-vocabulary understanding capabilities similar to the original VLM $\theta$. Consequently, adaptation performance needs to be measured over *base* queries $Q_{\text{base}}$ and *novel* queries $Q_{\text{novel}}$. *Base* queries in the test set also appear in the training queries $Q_{\text{train}}$ and are typically prevalent in the training set (*e.g.*, "car" in the urban driving domain), while *novel* queries have not been seen during adaptation.

## 4 METHOD

Following this task definition (Sec. 3), we present a method to adapt CLIP (or similar VLMs) to a specific target domain for open-vocabulary segmentation. We first explain the preliminaries required for our method (Sec. 4.1). Then we propose a simple yet effective way for prompt tuning (Sec. 4.2), and introduce a novel training strategy based on a triplet loss (Sec. 4.3). This training strategy is designed to maintain the structure of CLIP's embedding space while injecting domain-specific priors to the model. Finally, we discuss how to mine data for the triplet loss (Sec. 4.4).

### 4.1 PRELIMINARIES

The architecture of open-vocabulary segmentation pipelines typically comprises a) a class-agnostic mask proposal component that generates potential masks along with their visual embeddings, b) a pre-trained VLM text encoder to output text embeddings for each language query, and c) a pre-trained VLM image encoder that outputs visual embeddings given the mask proposals. Following most open-vocabulary segmentation models, in this paper, we use CLIP (Radford et al., 2021) with a Vision Transformer (ViT) backbone as VLM.

During inference, the relevance score between text queries and each mask proposal is computed as the cosine similarity between the corresponding visual embedding $\mathbf{v}$ and text embeddings

$\{\mathbf{t}_1, \cdots, \mathbf{t}_N\}$ of all $N$ queries. The query with the highest score corresponding to the semantic prediction $\hat{\mathbf{y}}$ for this mask, is denoted as: $\hat{\mathbf{y}} = \arg\max_n \left\{ \frac{\mathbf{v} \cdot \mathbf{t}_n}{\|\mathbf{v}\| \cdot \|\mathbf{t}_n\|} \right\}$

Despite promising results, open-vocabulary segmentation faces significant challenges, particularly due to the limited specialized domain knowledge of CLIP embeddings. These limitations hinder the segmentation precision across diverse domains (see Fig. 1). Drawing inspiration from the success of prompt tuning in enhancing classification accuracy across various domains (Zhou et al., 2022d;c; Jia et al., 2022; Khattak et al., 2023a; Liang et al., 2023), we explore its potential for our task.

In prompt tuning, learnable tokens are appended to the user-provided input query (usually text or image) of the model. Prompt tuning enables adapting CLIP to target domains by *learning* input prompts instead of handcrafting prompts. The additional appended tokens provide contextual information on target domains while freezing the original model parameters. This way, only a small fraction of new learnable parameters are added, making the learning process more efficient. Following these works, we propose a novel prompt tuning approach to specifically refine CLIP for improved domain-specific segmentation, as outlined next in Sec. 4.2

### 4.2 VISUAL AND TEXTUAL PROMPT TUNING

We first concatenate a set of learnable prompts to the image patch embeddings and text embeddings. After appending the prompt vectors, the enhanced tensors are formed for image visual embeddings, denoted as $\mathbf{v}^{(0)}$ and text embeddings, denoted as $\mathbf{t}^{(0)}$:

$$\mathbf{v}^{(0)} = [\mathrm{v}^{(0)}; \mathbf{e}_v^{(0)}; \mathbf{p}_v^{(0)}] \quad \text{and} \quad \mathbf{t}^{(0)} = [\mathrm{t}^{(0)}; \mathbf{e}_t^{(0)}; \mathbf{p}_t^{(0)}] \tag{1}$$

where $\mathrm{v}^{(0)}$ and $\mathrm{t}^{(0)}$ represent [CLS] and [EOS] special token embeddings, and $\mathbf{e}_v^{(0)}$ and $\mathbf{e}_t^{(0)}$ are the visual and text embeddings. As illustrated in Fig. 1 *(center)*, $\mathbf{p}_v^{(0)} = (\{p_k^v\}_{k=1}^K)^{(0)}$ and $\mathbf{p}_t^{(0)} = (\{p_k^t\}_{k=1}^K)^{(0)}$ correspond to the learnable prompts added in the input space, where $K$ is the total number of learnable prompts and $p_k^v, p_k^t$ are the $k$-th learnable prompt. Note that we initialize text prompts, $\mathbf{p}_t^{(0)}$ with the tokenization of "A photo of a" for the prompts added in the input space, while $\mathbf{p}_v^{(0)}$ is initialized from a random distribution (Khattak et al., 2023a).

Next, we append $K$ learnable prompts into deeper layers. That is, we define $\mathbf{v}^{(j)} = [\mathbf{e}_v^{(j)}; \mathbf{p}_v^{(j)}]$ and $\mathbf{t}^{(j)} = [\mathbf{e}_t^{(j)}; \mathbf{p}_t^{(j)}]$ where $\mathbf{v}^{(j)}, \mathbf{t}^{(j)}$ are the input tensors to the $(j+1)$-th layer, $1 \le j < J$ and $J$ is the prompt depth, *i.e.*, the model depth up to which learnable prompts are inserted. If $J=1$, we add prompts only to the input of the first hidden layer, and the model defaults to combining CoOp (Zhou et al., 2022d) for the text encoder and VPT-Shallow (Jia et al., 2022) for the visual encoder. $J$ is bounded by the number of layers of the visual/text encoders. If $J$ is smaller than the total number of layers, for the remaining encoder layers after the $J$-th layer, we feed the preceding layer's prompt embedding through the remaining layers (Khattak et al., 2023a; Lee et al., 2023).

### 4.3 OPTIMIZATION

Next, we introduce how to optimize the visual prompts $\mathbf{p}_v^{(j)}$ and text prompts $\mathbf{p}_t^{(j)}$ in each layer as discussed in Sec. 4.2. The goal is to adapt the model to target domain while maintaining the overall structure of the embedding space, crucial for open-vocabulary understanding. We first optimize only the visual prompts, then only the text prompts. This sequential approach is motivated by our experiments, indicating that the triplet loss with negative queries does not benefit the visual prompts and two-stage training can better preserve the alignment with the original CLIP embedding space.

**Optimization of Visual Prompts.** In each iteration, we randomly sample a batch of 16 image segments with their ground truth class names, passing through the CLIP visual and text encoder to obtain their embedding $\mathbf{v}_i$ and $\mathbf{t}_i$ for each segment $i$. We use a cross-entropy loss $\mathcal{L}_{ce}(\mathbf{v}_i, \mathbf{t}_i)$ to optimize the visual prompts $\mathbf{p}_v^{(j)}$ based on the computed logits within the label space $Q_{base} \cup Q_{negative}$, where $Q_{base}$ is the set of base queries introduced in the training set, and $Q_{negative}$ denotes the negative queries that are generated by GPT-4 to augment the label space (as introduced in Sec. 4.4).

**Optimization of Text Prompts.** Once optimized, we freeze the visual prompts and solely optimize the text prompts $\mathbf{p}_t^{(j)}$ with an objective $\mathcal{L}(\mathbf{v}_i, \mathbf{t}_i^+, \mathbf{t}_i^-)$, where $\mathbf{v}_i, \mathbf{t}_i^+$ and $\mathbf{t}_i^-$ are the visual embedding, true class name, and negative class name embeddings corresponding to the segment $i$, and $\mathcal{L}_t$ is a

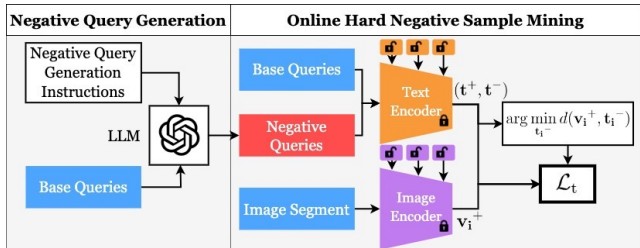

**Figure 4: Triplet Mining**. We first instruct GPT-4 (Achiam et al., 2023) to generate negative queries for a given set of base queries. During training, we feed the base queries and negative queries along with the image segments to the model. Then, we perform online hard negative sample mining, where we find the query with the minimum distance to the visual embedding of the corresponding segment.

triplet loss (Balntas et al., 2016). $\mathcal{L}_{\mathrm{ce}}$ is again computed from the logits within the label space $Q_{\mathrm{base}} \cup Q_{\mathrm{negative}}$.

$$\mathcal{L}(\mathbf{v}_i, \mathbf{t}_i^+, \mathbf{t}_i^-) = \mathcal{L}_{\mathrm{ce}}(\mathbf{v}_i, \mathbf{t}_i^+) + \lambda \mathcal{L}_{\mathrm{t}}(\mathbf{v}_i, \mathbf{t}_i^+, \mathbf{t}_i^-) \tag{2}$$

$$\mathcal{L}_{\mathrm{t}}(\mathbf{v}_i, \mathbf{t}_i^+, \mathbf{t}_i^-) = \max\{\|\mathbf{v}_i - \mathbf{t}_i^+\|_2 - \|\mathbf{v}_i - \mathbf{t}_i^-\|_2 + \mu, 0\} \tag{3}$$

and the margin $\mu$ is set to 1.5 as higher margin can enhance base class separation but reduce generalization after a certain threshold (see Appendix F). To maintain the structure of the CLIP embedding space and preserve open-vocabulary understanding while adapting to a specific domain, we apply a triplet loss inspired by the contrastive objective from CLIP (Radford et al., 2021). This loss ensures that the embeddings of similar queries remain close together while pushing dissimilar ones apart, effectively retaining the original shape of the CLIP embedding space and its capacity for open-vocabulary comprehension. Note that, we gradually increase the $\lambda$ in Eq. 2 from $\lambda_{\min}$ to $\lambda_{\max}$.

### 4.4 TRIPLET MINING

**Negative Queries.** One of the challenges of employing triplet loss is to find proper negative samples to form triplets. Using triplets with randomly selected negative samples will get the optimization stagnate quickly (Hermans et al., 2017). To address this, we instruct GPT-4 (Achiam et al., 2023) to generate 5 similar queries for each base query as shown in Fig. 4 *(left)*. These text phrases should be challenging for a machine learning model to differentiate, yet easily distinguishable by humans. For instance, "*ceiling*" and "*ceiling fan*" are hard to distinguish for models but have a semantic difference. In Appendix C, we list the detailed GPT-4 instructions to generate negative samples.

**Online Hard Negative Sample Mining.** After generating the negative queries, we employ an online hard negative mining strategy to identify informative triplets with the hardest negatives during training, as outlined in (Hermans et al., 2017; Schroff et al., 2015; Simo-Serra et al., 2015; Xuan et al., 2020) (Fig. 4, *right*). This approach is crucial for enhancing the model's ability to differentiate between similar yet distinct classes, thereby increasing its precision. For a dataset with $N$ classes, we find the hardest negative query for each segment on the fly from the remaining $N-1$ classes and the generated negative queries. In particular, we find the hardest negative by the lowest $L_2$ distance between its text embedding and the visual embedding $\mathbf{v}_i$ of the target segment $i$. Combined with the segment's true class name and visual embedding, it forms the triplet to refine the text prompts.

### 4.5 APPLICATION IN EXISTING 2D & 3D OPEN-VOCABULARY SEGMENTATION PIPELINES

After training our OpenDAS, we apply it to existing open-vocabulary segmentation (OVS) pipelines, namely FC-Clip (Yu et al., 2024) for 2D images, and OpenMask3D (Takmaz et al., 2023a) for 3D point clouds. Both follow the common architecture as defined in Sec. 4.1 with a mask proposal generator followed by an open-vocabulary segment classification module. Our method can be integrated as a plug-and-play component into these methods, replacing the original image and text encoders.

## 5 EXPERIMENTS

As defined in Sec. 3, domain adaptation for OVS requires different data for testing than for adaptation training to evaluate novel classes not seen during adaptation. To that end, the test queries are split into *base queries* that are also present during the adaptation, and *novel queries* that test the open-vocabulary understanding of our adapted model. The experiments are performed on three datasets, covering indoor and outdoor domains.

**ADE20K-150.** (Zhou et al., 2017; 2019) covers indoor and outdoor scenes with 2000 images for validation and 150 distinct classes. It is widely used to evaluate OVS models (Cho et al., 2023; Xu et al., 2023b; Yu et al., 2024; Liang et al., 2023; Naeem et al., 2023; Xie et al., 2023).

**KITTI-360.** (Liao et al., 2022) covers urban driving scenes with 37 semantic labels. For training, we use the annotated 2D images from the training split and evaluate it on the validation split.

**ScanNet++ Offices.** (Yeshwanth et al., 2023) consists of 3D reconstructions of over 450 indoor scenes including iPhone RGB-D streams. To test the generalization to novel queries, we construct a subset of ScanNet++. We refer to this subset as ScanNet++ Offices. For this, we visually identify 30 office scenes, covering various university rooms. Then, we split the subset to test our model's open-vocabulary classification capabilities. Specifically, we use 14 scenes (7989 images) for adaptation training and test on 16 scenes (11054 images). The queries are split into 156 training and 233 test labels. Of those, 108 are present in both sets. This allows us to test with 108 base and 125 novel queries. Please refer to Appendix E for the scene IDs used for training and testing.

**Baselines.** We compare our approach to state-of-the-art prompt learning methods CoCoOp (Zhou et al., 2022c), VPT (Jia et al., 2022), RPO (Lee et al., 2023), and MaPLe (Khattak et al., 2023a). We investigate these methods for segment classification for the first time. Further comparisons against the WiSE-FT robust fine-tuning method (Wortsman et al., 2022) can be found in Appendix B.

**Metrics.** To measure the adaptation performance with ground-truth masks, we employ established metrics for image classification, *i.e.*, Accuracy (Acc), and a family of F1 scores, including Weighted-F1 (W-F1) by weighing the number of occurrences for each label, Base-F1 (B-F1) over base queries that are seen during adaptation training, and Novel-F1 (N-F1) over novel queries. In experiments with predicted masks, we measure the commonly used metrics mean IoU (mIoU) and the frequency weighted IoU (fwIoU) for 2D OVS and AP (Average Precision), $AP_{50}$ and $AP_{25}$ for 3D OVS tasks.

**Implementation Details.** Following previous prompt learning approaches (Zhou et al., 2022d;c; Khattak et al., 2023a; Lee et al., 2023), we use the Dassl library (Zhou et al., 2021; 2022b) to implement prompt tuning on CLIP with the triplet loss. We first optimize only the visual prompts for 5 epochs. During training, we have a warmup epoch with a learning rate of $10^{-5}$ and then set the base learning rate to 0.0025 with a cosine scheduler from the second epoch. After training visual prompts, we only optimize the text prompts for another 5 epochs with the same base learning rate. We use a batch size of 16 and an SGD optimizer on a single NVIDIA A100 GPU, and the training time is around 10-15 hours in total depending on the dataset. The 2D segments excerpted from different datasets have filled background with the average pixel value of CLIP training images as done by Liang et al. (2023). The $\lambda_{min}$ and $\lambda_{max}$ is set to be 2 and 5, respectively.

## 5.1 MAIN RESULTS

**Adaptation Performance on Base Queries.** In this first experiment (Tab. 1), we evaluate the effect of prompt tuning for domain adaptation. We report experimental results on two *closed-set* datasets, KITTI-360 and ADE20K. This setup follows the existing adaption procedure where the VLM is trained and tested on the same queries, *i.e.*, $Q_{test} = Q_{train} = Q_{base}$. For each method, the image and ground-truth segmentation masks are presented and we measure how well the VLM matches the segments to the queries. In general, prompt learning techniques significantly improve CLIP's segment classification capabilities in the target domain: all adapted models improve over the original CLIP. The experiment also reveals that deep multimodal prompt tuning approaches, MaPLe and OpenDAS, improve upon both unimodal approaches, CoCoOp and VPT, and shallow multimodal approach, RPO, in the supervised domain adaptation setting. Our approach improves over all adaptation methods by a significant margin while using only a fraction (1.2%) of the number of parameters of the second best-performing method MaPLe. This is due to the two-stage training setting preserving the semantic alignment of text features with the adapted visual features while enabling full control over the adaptation of each modality.

**Open-Vocabulary Understanding for Segmentation.** A critical aspect of domain adaptation is that it might negatively affect the open-vocabulary capabilities of VLMs. To evaluate the adapted models on novel classes, we consider two test cases (see Tab. 2). Our curated SN++ Offices already has 125 novel queries in the test set. Furthermore, we test a model adapted on ADE20K, a dataset spanning indoors and urban outdoors, on SN++ Offices (indoors) and KITTI-360 (urban driving). Because the domains overlap, but the annotated categories do not fully match between the datasets,

| Adaptation Method | Modality | #Params. | KITTI-360 Acc. | KITTI-360 W-F1 | ADE20K-150 Acc. | ADE20K-150 W-F1 |
|---|---|---|---|---|---|---|
| No adaptation | | 0 | 19.1 | 23.4 | 27.8 | 32.7 |
| CoCoOp (Zhou et al., 2022c) | 📘 | ∼ 77K | 61.1 | 59.7 | 54.2 | 51.8 |
| VPT (Jia et al., 2022) | 🎨 | ∼ 786K | 65.2 | 67.7 | 58.2 | 59.8 |
| RPO (Lee et al., 2023) | 🎨📘 | ∼ 43K | 66.0 | 63.6 | 58.1 | 55.2 |
| MaPLe (Khattak et al., 2023a) | 🎨📘 | ∼ 18935K | 69.9 | 68.6 | 67.7 | 65.9 |
| OpenDAS (**Ours**) | 🎨📘 | ∼ 233K | **75.7**(+5.8) | **75.2**(+6.6) | **73.1**(+5.4) | **71.9**(+6.0) |

**Table 1: Adaptation Performance on Base Queries.** We compare different adaptation methods on outdoor data (KITTI-360) and a combination of both indoor and outdoor data (ADE20K-150). Some methods adapt only the text-encoder (📘), only the image-encoder (🎨), or the encoders for both modalities (🎨 📘). We also report the number of additional trainable parameters introduced by the adaptation method (#Params). In these experiments, queries during adaptation training and test time are the same, *i.e.*, $Q_{test} = Q_{train} = Q_{base}$.

| Method | Modality | # Params | SN++ Offices W-F1 | SN++ Offices B-F1 | SN++ Offices N-F1 | ADE20K → SN++ Offices W-F1 | ADE20K → SN++ Offices B-F1 | ADE20K → SN++ Offices N-F1 | ADE20K → KITTI-360 W-F1 | ADE20K → KITTI-360 B-F1 | ADE20K → KITTI-360 N-F1 |
|---|---|---|---|---|---|---|---|---|---|---|---|
| No adaptation | | 0 | 11.2 | 11.0 | 12.0 | 11.2 | 11.3 | 11.0 | 24.1 | 23.0 | 24.9 |
| CoCoOp (Zhou et al., 2022c) | 📘 | ∼ 77K | 25.7 | 34.3 | 12.7 | 11.2 | 18.0 | 9.9(-1.1) | 27.1 | 30.4 | 22.1(-2.8) |
| VPT (Jia et al., 2022) | 🎨 | ∼ 786K | 33.8 | 37.6 | 12.8 | 13.0 | 19.2 | 8.8(-2.2) | 29.5 | 34.8 | 25.8 |
| RPO (Lee et al., 2023) | 🎨📘 | ∼ 43K | 30.6 | 40.9 | 14.9 | 13.4 | 13.9 | 13.3 | 33.7 | 42.8 | 19.9(-5.0) |
| MaPLe (Khattak et al., 2023a) | 🎨📘 | ∼ 18935K | 36.3 | 48.1 | 18.4 | 18.8 | 29.3 | 16.8 | 43.5 | 57.7 | 22.2(-2.7) |
| OpenDAS (**Ours**) | 🎨📘 | ∼ 233K | **40.2** | **51.5** | **23.0**(+4.6) | **23.0** | **30.4** | **21.6**(+4.8) | **47.1** | **60.8** | **26.6**(+4.4) |

**Table 2: Open-Vocabulary Understanding for Segmentation.** We evaluate segmentation over base queries that have also been part of the adaptation training (B-F1) as well as a generalization to novel queries (N-F1) and the overall weighted F1 (W-F1). To be able to test on novel queries, we evaluate on ScanNet++ Offices (SN++ Offices) and cross-dataset by adapting to ADE20K-150 (ADE20K) and testing on ScanNet++ Offices and KITTI-360. Performance degradation compared to the original CLIP baseline is shown in red.

| Method | ADE20K-150 mIoU (%) | ADE20K-150 fwIoU (%) |
|---|---|---|
| OVSeg | 29.8 | 57.8 |
| + OpenDAS | 35.8 (+6.0) | 64.3 (+6.5) |
| FC-CLIP | 34.3 | 59.9 |
| + OpenDAS | 37.3 (+3.0) | 64.7 (+4.8) |

| Method | ScanNet++ Offices AP | ScanNet++ Offices AP$_{50}$ | ScanNet++ Offices AP$_{25}$ |
|---|---|---|---|
| OpenMask3D | 8.1 | 11.5 | 14.1 |
| + OpenDAS | **12.2** (+4.1) | **18.0** (+6.5) | **24.0** (+ 9.9) |

**Table 3: Segmentation Performance with Predicted Segments.** We apply our method to recent open-vocabulary 2D semantic segmentation models OVSeg (Liang et al., 2023), FC-CLIP (Yu et al., 2024) and SOTA open-vocabulary 3D instance segmentation model OpenMask3D (Takmaz et al., 2023a).

this allows us to test 18 base and 19 novel queries in KITTI-360 and 47 base and 186 novel queries in SN++ Offices. As expected, the original CLIP performs equally well in base and novel classes. Existing adaption methods exhibit noticeable performance boosts over the original CLIP on base classes (W-F1 and B-F1 scores), but improve only marginally on novel classes (N-F1 scores). In contrast, OpenDAS demonstrates superior performance in both base and novel classes. Similarly, when trained on ADE20K-150 and evaluated cross-dataset, we observe significant improvements over all baselines, especially for novel classes where OpenDAS even achieves higher N-F1 than the original CLIP. In contrast, other methods occasionally show a decrease in open-vocabulary generalization following adaptation. This suggests that the triplet loss effectively preserves the structured CLIP embedding space while the adaptation process closes the domain gap between CLIP's training images and the target data. We provide further analysis in Appendix G.

**Segmentation Performance with Predicted Segments.** Besides evaluating our segment classification performance given ground truth masks, we additionally apply our prompt tuning method predicted masks from OVSeg (Liang et al., 2023), FC-CLIP (Yu et al., 2024) and OpenMask3D (Takmaz et al., 2023a), assessing whether our method can help them better understand the semantics of their predicted segments. As shown in Tab. 3, we observe the performance boost in all metrics. Our method shows especially significant improvements with lower IoU thresholds than the original OpenMask3D. This shows that OpenDAS can be directly incorporated into existing OVS pipelines and improve their performance. Further comparisons with predicted masks can be found in Appendix H.

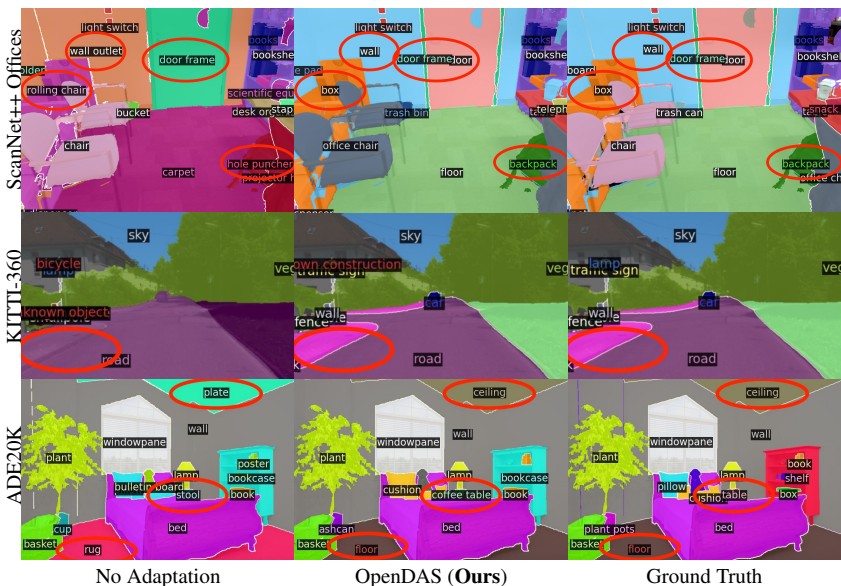

**Figure 5: Qualitative Results on 2D Segment Classification.** We show the predicted object classes with the ground truth masks given on three datasets.

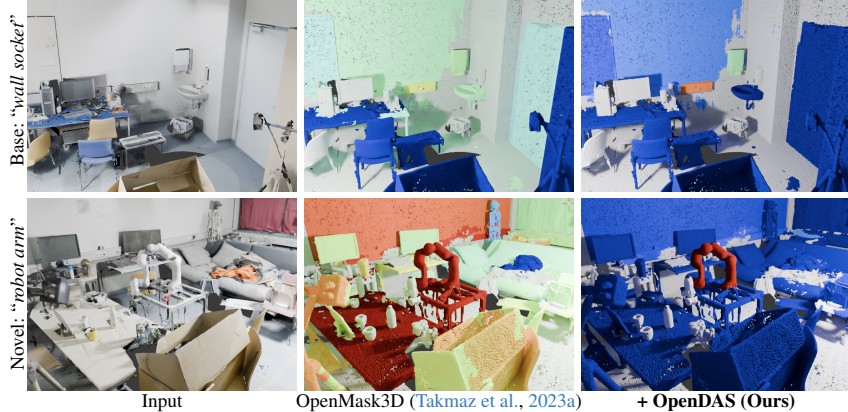

**Figure 6: Qualitative Results on Open-Vocabulary 3D Instance Segmentation.** We show the query-response scores of OpenMask3D (Takmaz et al., 2023a) predicted with CLIP (Radford et al., 2021) and our OpenDAS. Blue indicates low similarity with the text query and red high similarity. Unlike CLIP, our Open-DAS can better localize the correct mask with a higher similarity score.

**Qualitative Results.** Fig. 5 shows qualitative results on ScanNet++ Offices (Yeshwanth et al., 2023), KITTI-360 (Liao et al., 2022), and ADE20K-150 (Zhou et al., 2017; 2019). Our method shows clear improvements over CLIP, distinguishing classes like "door frame"–"door", "road"–"sidewalk", "plate"–"ceiling". Fig. 6 demonstrates how OpenDAS can boost the performance of OpenMask3D (Takmaz et al., 2023a) by replacing the original CLIP with our adapted VLM. The improvement in seen classes like "wall socket" is anticipated. However, as OpenDAS is adapted to predict from masked images, it also improves predictions on novel queries like "robot arm".

### 5.2 ABLATION STUDIES

**What is the influence of $\lambda_{max}$?** Setting $\lambda_{max} = 0$ indicates that the training objective in Eq. 2 for text prompts defaults to only cross-entropy loss over the base and negative queries as the objective, while ignoring the triplet loss. As $\lambda_{max}$ increases, the weight $\lambda$ for triplet loss also increases during training. Tab. 4 shows that the performance has a peak when we set $\lambda_{max} = 5$ for both datasets.

**Joint *vs.* Sequential Training Setting.** Lee et al. (2023) and Khattak et al. (2023a) suggest that the image and text encoders should be trained jointly. Our ablation, however, shows that a two-stage

| $\lambda_{\max}$ | SN++ Offices Acc | W-F1 | KITTI-360 Acc | W-F1 | ADE20K-150 Acc | W-F1 |
|---|---|---|---|---|---|---|
| 0 | 38.9 | 34.8 | 66.4 | 64.4 | 51.7 | 48.4 |
| 2 | 39.8 | 35.8 | 71.9 | 70.2 | 55.8 | 54.0 |
| **5** | **40.1** | **36.5** | **72.9** | **70.9** | **65.7** | **63.6** |
| 10 | 39.3 | 35.7 | 72.1 | 70.8 | 58.9 | 56.5 |

Table 4: **What is the influence of $\lambda_{\max}$?** Acc and W-F1 on ScanNet++ (SN++) Offices, KITTI-360, and ADE20K-150. When $\lambda_{\max} = 0$, we default to using only cross-entropy loss over base and negative queries.

| Training Setting Stage 1 | Stage 2 | SN++ Offices Acc | W-F1 | KITTI-360 Acc | W-F1 | ADE20K-150 Acc | W-F1 |
|---|---|---|---|---|---|---|---|
| Joint + T | | 39.9 | 36.2 | 72.3 | 71.2 | 68.8 | 67.2 |
| 🟦 + T | 🍪 | 33.9 | 29.9 | 72.5 | 71.2 | 59.4 | 57.1 |
| 🍪 + T | 🟦 + T | 42.3 | 38.8 | 71.9 | 71.7 | **73.3** | **72.1** |
| 🍪 | 🟦 + T | **43.7** | **40.2** | **75.7** | **75.2** | 73.1 | 71.9 |

Table 5: **Joint vs. Sequential Training Setting.** Comparison of training settings: joint training with triplet loss (+ T) and two-stage settings: (🟦 + T, 🍪), (🍪 + T, 🟦 + T), and (🍪, 🟦 + T). Acc and W-F1 are measured on ScanNet++ (SN++) Offices, KITTI-360, and ADE20K-150.

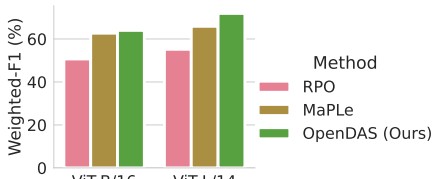

Figure 7: **Ablation on ViT Backbone on ADE20K-150** Zhou et al. (2017; 2019). Our method is robust to different backbones, ViT-B/16 (86M parameters) and ViT-L/14 (307M parameters).

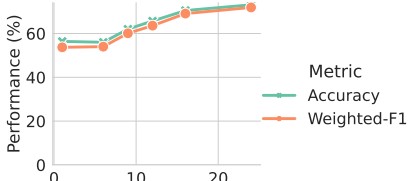

Figure 8: **Up to what model depth $J$ should prompts be added?** We show the effect of $J$ on ADE20K. Adding prompts to more layers improves the performance of OpenDAS.

training of first visual prompt optimization (🍪), followed by textual prompt optimization with triplet loss (🟦 + T) achieves improved performance, as the two-stage training preserves the alignment of adapted CLIP image encoder to the original text encodings and triplet loss with negative queries does not boost the performance when it is applied to the visual prompts (see Tab. 5).

**ViT Backbone.** We present a comparison of the visual backbone and its impact on the Weighted-F1 score in Fig. 7. The default setting for OVSeg (Liang et al., 2023) and OpenMask3D (Takmaz et al., 2023a) is based on ViT-L/14. Thus, we use ViT-L/14 for all our experiments, unlike prior prompt learning methods that base all their experiments with ViT-B/16. However, our method demonstrates robustness across different backbones, showing performance improvements even with the smaller backbone. Additional ablation studies on more datasets are provided in Appendix F.

**Up to what model depth $J$ should prompts be added?** If $J = 1$, we add learnable prompts only to the input space. With increasing $J$, we add learnable prompts to more layers, up to $J = 24$, where we add prompts to all layers. In Fig. 8, we observe that OpenDAS improves with an increase of $J$. We refer the readers to Appendix F for the ablations on other datasets.

## 6 CONCLUSION & LIMITATIONS

We introduce a new task "open-vocabulary domain adaptation". We focus on segmentation tasks and explore prompt tuning for domain adaptation while keeping generalization to novel queries. We show that existing prompt tuning methods, especially when combined with triplet loss and auxiliary negative queries, can significantly enhance VLMs' performance for open-vocabulary segmentation tasks in a parameter-efficient way. Our two-stage training scheme with triplet loss improves adaptation, achieving better results in both base and novel classes. Applying our model with ground-truth masks to different datasets yields significant improvements over previous methods, demonstrating the efficacy of OpenDAS. We also show that integrating our adapted model into existing OVS pipelines boosts performance in both 2D and 3D OVS tasks.

Despite promising results, our work has limitations. All evaluated methods require annotated ground-truth segmentation, which is expensive to obtain. Future work could explore few-shot learning settings for OVDA, increasing practicality for real-world applications. Finally, our experiments were limited as we only consider a decoupled setting for mask proposal generation and class prediction. Adaptation methods for coupled OVS models could be investigated.

## 6.1 Reproducibility Statement

For the reproducibility of our experiments, we share our code and implementation details in the supplementary material. The code provided involves the method described in Sec. 4 including the integration to the existing pipelines OVSeg and OpenMask3D. For the implementation details, we refer the readers to Appendix D. As we create our own split from ScanNet++ for some experiments, we also share the chosen scene IDs in Appendix E.

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

## A  FURTHER QUALITATIVE RESULTS AGAINST PRIOR PROMPT TUNING METHODS

In the provided figures (see Fig. 9, Fig. 10, Fig. 11), we compare our method's segment classification capabilities over the other baseline methods, as well as the ground truth labels for reference. Fig. 9 presents a qualitative comparison on ScanNet++ (Yeshwanth et al., 2023). Our method, OpenDAS, displays robustness in classifying basic elements like "wall", "floor", and "whiteboard" across various viewpoints. OpenDAS adeptly differentiates between conceptually similar objects, for instance, "ceiling" - "ceiling beam" and "wall" - "objects" – a distinction that poses a challenge for other methods and is likely encouraged by our triplet loss. However, some failures remain, e.g., "object" instead of "kettle" in the first column, and instead of "window frame" in the second column.

| Method | Modality | # Params | time/iter | SN++ Office W-F1 | B-F1 | N-F1 | KITTI-360 Acc | W-F1 | ADE20K-150 Acc | W-F1 |
|---|---|---|---|---|---|---|---|---|---|---|
| No adaptation | | 0 | | 11.2 | 11.0 | 12.0 | 19.1 | 23.4 | 27.8 | 32.7 |
| WiSE-FT Wortsman et al. (2022) | 📘 | ∼ 123M | 0.85 s | 31.0 | 35.0 | 29.5 | 53.9 | 58.8 | 47.9 | 51.0 |
| WiSE-FT Wortsman et al. (2022) | 🎨 | ∼ 304M | 1.05 s | **45.9** | 47.3 | **45.3** | **78.8** | **80.5** | **73.9** | **74.8** |
| OpenDAS (**Ours**) | 🎨📘 | ∼ 233K | **0.53 s** | 40.2 | **51.5** | 23.0 | 75.7 | 75.2 | 73.1 | 71.9 |

**Table 6: Comparison with Robust Fine-Tuning Wortsman et al. (2022) on ScanNet++ Office, KITTI-360, and ADE20K-150.** We can see that we significantly outperform the text-encoder fine-tuning setting on all three datasets with about $1000\times$ fewer parameters. We also present competitive results over their image-encoder fine-tuning setting that uses over $2000\times$ more parameters, and show stronger B-F1 results on ScanNet++, while being $2\times$ faster to train.

For KITTI-360 (Liao et al., 2022), we present comparisons in Fig. 10. The task is relatively simpler as we have only 37 semantic classes. Among these, several adapted models struggle to separate "road" from "sidewalk", especially in instances where they share similar coloration. OpenDAS, leveraging the nuanced capabilities provided by triplet loss during training, successfully identifies and segregates these analogous classes.

In Fig. 11, we present a qualitative comparison on the ADE20K-150 dataset (Zhou et al., 2019; 2017). As the task is closed-set with 150 classes, all prompt learning methods perform generally accurately on this dataset. However, we see that in some cases, multi-modal prompt tuning as done in RPO (Lee et al., 2023) and MaPLe (Khattak et al., 2023a) can result in a degradation in CLIP's original representations, leading to occasional misclassification between "sky" and other entities, unlike VPT (Jia et al., 2022) and OpenDAS, which employ isolated visual prompt tuning. Similarly, we observe some other failures with distinguishing "table" and "chair" in the second column by other methods and "grass" from other classes. OpenDAS, while generally proficient, is not without its faults, as evidenced by the occasional inability to discriminate "grass" from "earth" or the misidentification of a glass door as a "mirror".

## B  COMPARISON AGAINST ROBUST FINE-TUNING METHOD WISE-FT (WORTSMAN ET AL., 2022)

We further compare our approach against a robust fine-tuning method that fine-tunes the entire CLIP text and visual encoder, respectively. It differs from standard fine-tuning as it ensembles the weights of pre-trained and fine-tuned model weights to keep the pre-trained model's generalization capabilities. We show comparisons in Table 6 on KITTI-360 and ADE20K on the closed-vocabulary setting, as well as ScanNet++ Offices on the open-vocabulary setting. When compared to fine-tuning the CLIP text encoder, we achieve significant improvements in all metrics and all three datasets by only training ∼0.1% of CLIP-text encoder parameters. Looking into the comparison over fine-tuning the visual encoder, we show competitive results with only ∼0.05% of the parameters and achieve significant improvements on base classes in the ScanNet++ Office dataset.

## C  NEGATIVE QUERIES FOR TRIPLET LOSS

Our preliminary analysis indicates that when triplet loss is trained with easy negatives, the learned latent space cannot maintain the structure of the original embedding space, not generalizing well to unseen classes. Prior works on triplet loss (Hermans et al., 2017; Xuan et al., 2020) also indicate that hard negatives are essential to learning meaningful representations using triplet loss. By creating a negative query database, we augment the set of negative classes to distinguish from. For this purpose, we use GPT-4 (Achiam et al., 2023) to generate 5 negatives for each class. We give the following instructions:

"Your task is to produce five distinct examples for each class provided in the list, ensuring that the examples are not subcategories of each other but rather represent clear and separate entities within the same class. This means that each example should not be a subset or type of another example within the same category. The objective is to create similar examples that might be confused by

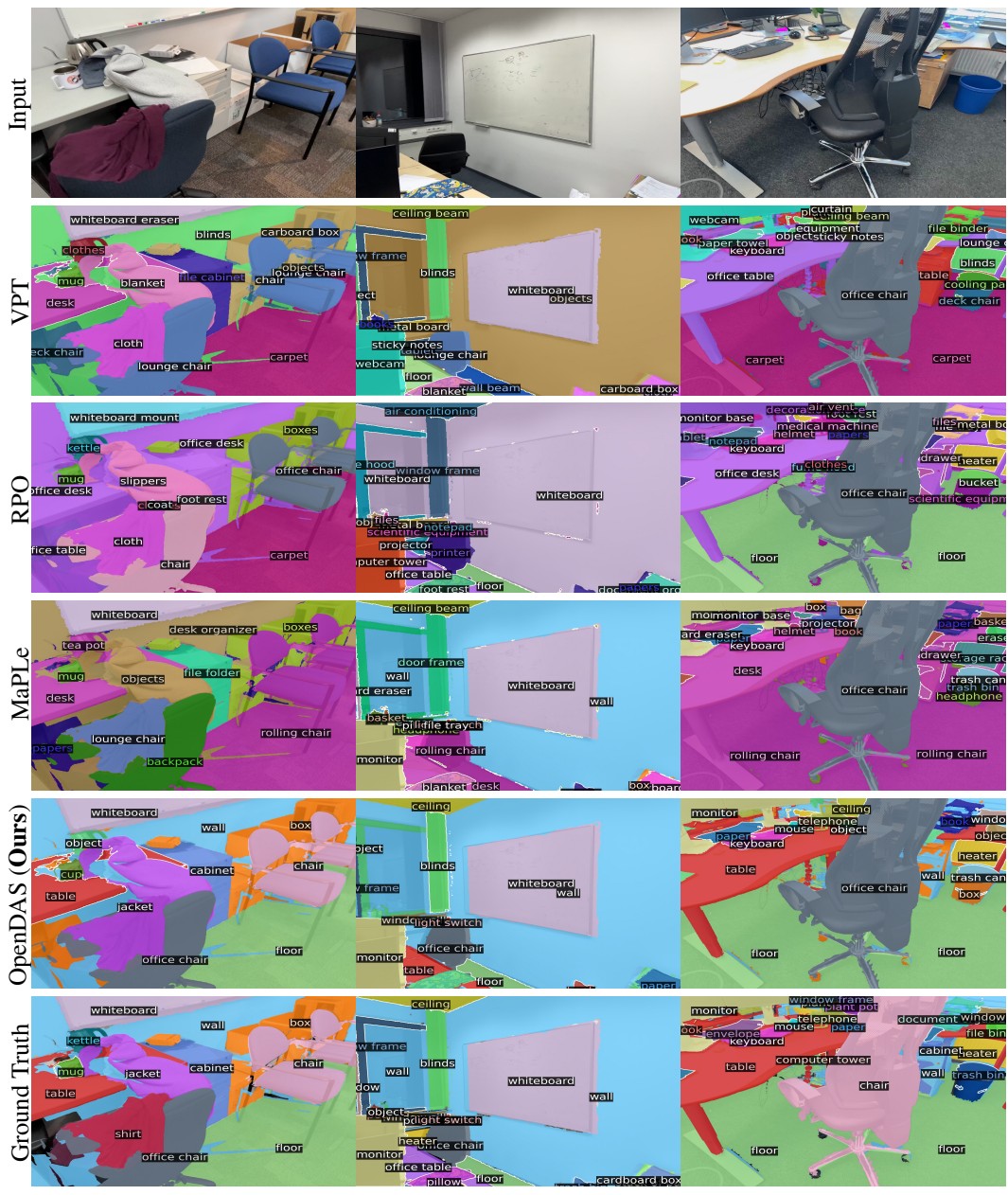

**Figure 9: Qualitative Comparison on Segment Classification on ScanNet++ Offices** Yeshwanth et al. (2023). We show the object classes with the ground truth masks predicted by baselines and OpenDAS, and ground truth labels. The masks are colorized based on the class ID. On the contrary to existing methods, our model can give the closest match to the ground truth labels exhibiting a similar color pattern.

a machine learning model but remain discernible to a human observer to be used as clear negative examples for triplet loss training. The output format should be a Python dictionary for easy integration."

Some examples of the generated classes are as follows.

- "wall": ["room divider", "partition", "divider screen", "privacy screen", "decorative panel"]

- "ceiling": ["chandelier", "pendant light", "skylight", "light fixture", "ceiling fan"]

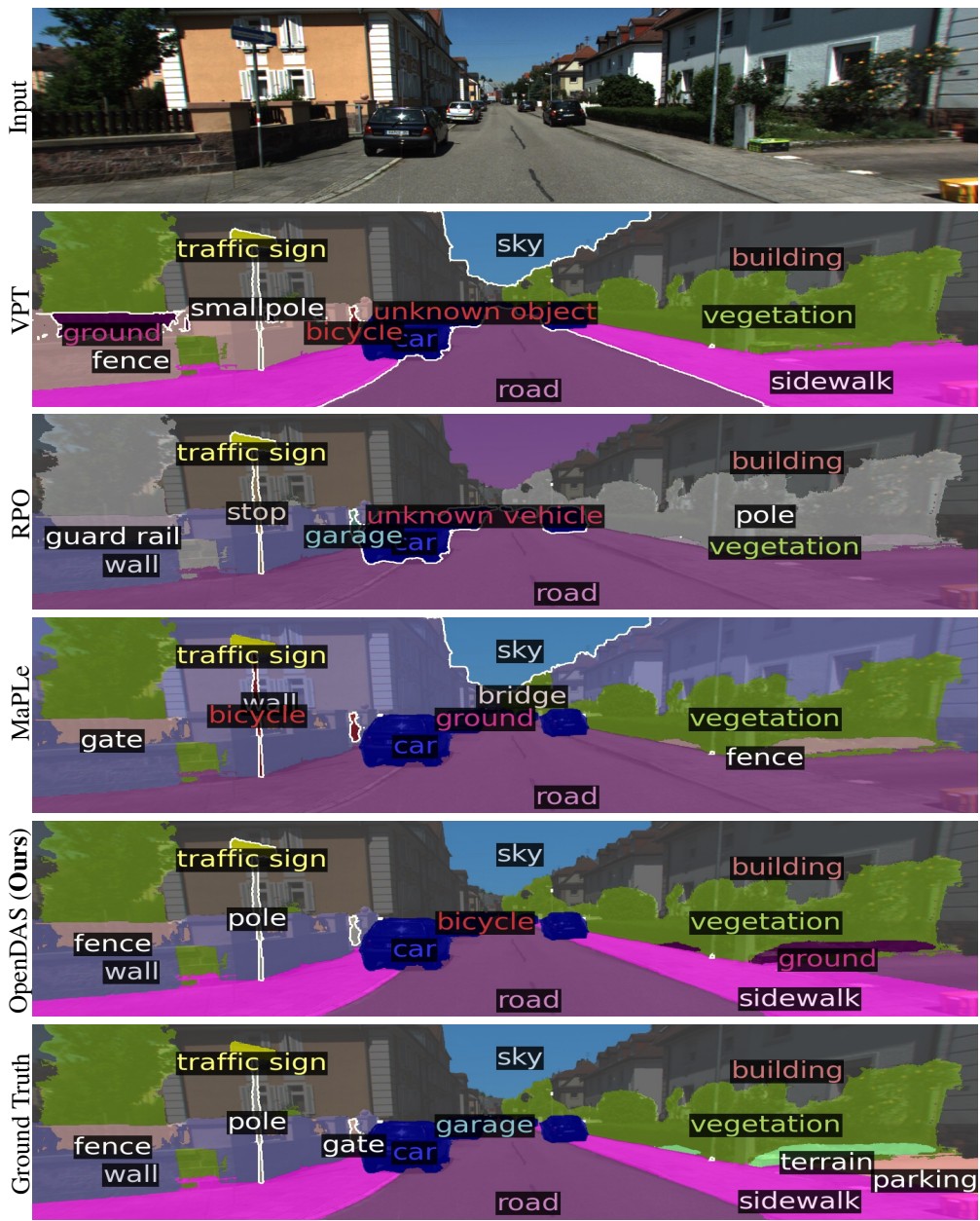

Figure 10: **Qualitative Comparison on Segment Classification on KITTI-360 Dataset** Liao et al. (2022). We show the predicted object classes with the ground truth masks given on three datasets. The masks are colorized based on the class ID. Unlike the existing methods, our model can give the closest match to the ground truth labels understanding the distinction between 'road' and 'sidewalk'.

• "folder organizer": ["bedside table", "end table", "chest of drawers", "bar stool", "storage ottoman"]

During training, we choose the hardest negative for each sample among all the training and negative classes combined to optimize the triplet loss.

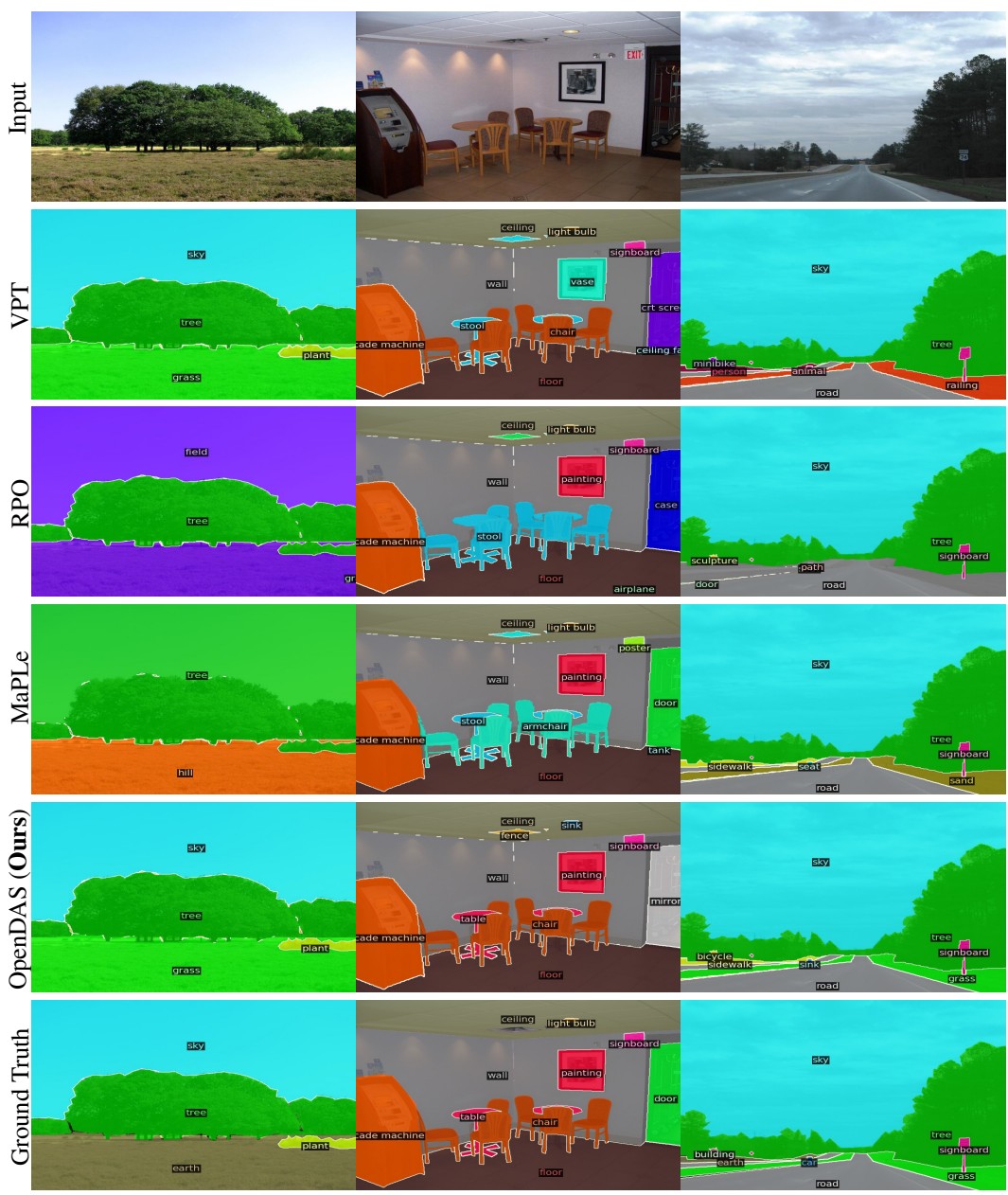

**Figure 11: Segment Classification Qualitative Comparison on ADE20K Dataset** Zhou et al. (2019; 2017).
We show the object classes with the ground truth masks predicted by baselines and OpenDAS, as well as
ground truth labels. The masks are colorized based on the class ID. We observe some improvements in the
classification, exhibiting closer color patterns to the Ground Truth labels compared to other baselines.

## D   FURTHER IMPLEMENTATION DETAILS

For the OpenDAS training pipeline, we first prepare segmentation datasets for the training of seg-
ment classification. Assuming that we have the 2D images as well as semantic annotations, we
perform pre-processing on the dataset to adapt the semantic annotations to the classification task.
As illustrated in figure 12, ground truth segmentation masks are applied, and the background is filled
with the mean pixel values from CLIP's original training images, mirroring the approach adopted by
a prior open-vocabulary segmentation work, OVSeg (Liang et al., 2023). Each segment is annotated
with a unique ID to facilitate the classification task.

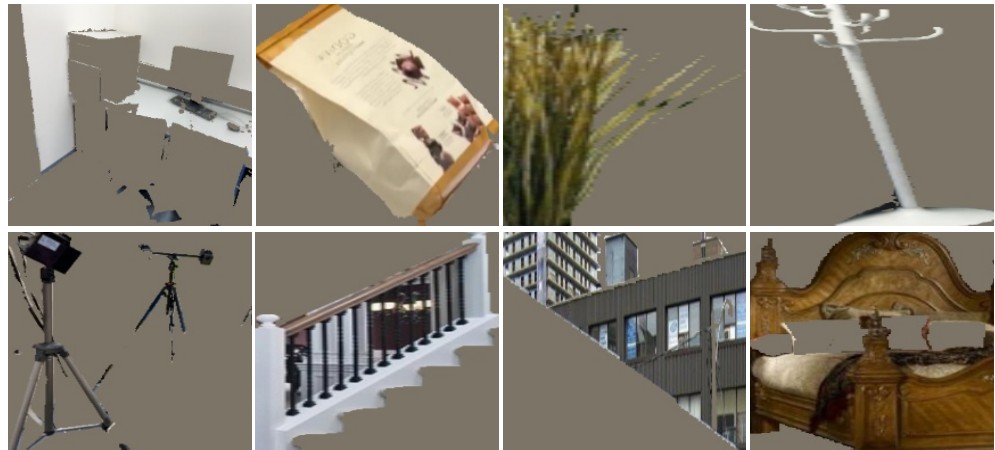

**Figure 12: Training Images.** We prepare classification annotations for the segment classification task by applying ground truth masks on images, where the background is filled with the mean pixel values from the original CLIP training images. Each segment is annotated with a unique ID for the classification task.

Having prepared the training set, we employ the Dassl library (Zhou et al., 2021; 2022b) to train our prompt learning approach, adhering to the conventions established by prior prompt learning methodologies (Zhou et al., 2022d;c; Khattak et al., 2023a; Lee et al., 2023). This way, we ensure compatibility with other baselines. We perform the inference with the same library and report the classification results measured with this implementation.

For class prediction with ground truth masks on a given image, we seamlessly integrate our tailored CLIP model with learned prompts into a segmentation framework built upon Detectron2 (Wu et al., 2019). This segmentation pipeline is based on OVSeg (Liang et al., 2023), thereby enabling the integration of our customized CLIP model into the OVSeg pipeline. This integration allows us to use class-agnostic mask predictions that come from MaskFormer (Cheng et al., 2021), as well as ground truth masks.

# E    SCANNET++ OFFICES

In this section, we provide more details about the ScanNet++ Office scenes, with 14 scenes for training and 16 for testing.

- Training scenes: ["0b031f3119", "1204e08f17", "260fa55d50", "394a542a19", "39f36da05b", "40b56bf310", "4ba22fa7e4", "75d29d69b8", "8b5caf3398", "1366d5ae89", "1a8e0d78c0", "2a496183e1", "30f4a2b44d", "419cbe7c11"]

- Test scenes: ["4a1a3a7dc5", "56a0ec536c", "59e3f1ea37", "7cd2ac43b4", "8d563fc2cc", "8e00ac7f59", "98b4ec142f", "9b74afd2d2", "9f139a318d", "e91722b5a3", "94ee15e8ba", "07f5b601ee", "2e74812d00", "036bce3393", "260db9cf5a", "28a9ee4557"]

# F    FURTHER ABLATION STUDIES

$\mu$ **Ablation.** In Tab. 7, we observe the impact of margin for the triplet loss. We see that if we evaluate the model closed-set, the larger margin is better for the model to distinguish base classes from each other easily. However, when we test the model on other datasets with novel queries, we see that after $\mu > 1.5$, the model's performance drops heavily on both ScanNet++ (SN++) Offices and KITTI-360. This is likely due to the larger margin causing the embeddings of novel classes to be pushed too far apart, leading to poor generalization for unseen queries.

| $\mu$ | ADE20K-150 | | ADE20K $\rightarrow$ SN++ Offices | | ADE20K $\rightarrow$ KITTI-360 | |
|---|---|---|---|---|---|---|
| | Acc | W-F1 | Acc | W-F1 | Acc | W-F1 |
| 1.0 | 72.8 | 71.3 | **24.5** | 20.1 | 41.7 | 36.6 |
| 1.5 | 73.1 | 71.9 | 22.8 | **23.0** | **47.3** | **47.1** |
| 2.0 | **73.8** | **72.5** | 19.9 | 12.9 | 41.7 | 35.8 |

**Table 7: $\mu$ Ablation.** We compare the effect of margin, $\mu$, to observe how the distance between anchor and the positive and negative labels affect the performance. We observe that if we only evaluate the model on base classes, higher $\mu$ gives better results. However, for novel classes, $\mu = 1.5$ gives the best performance. Hence, we apply this as the standard setting.

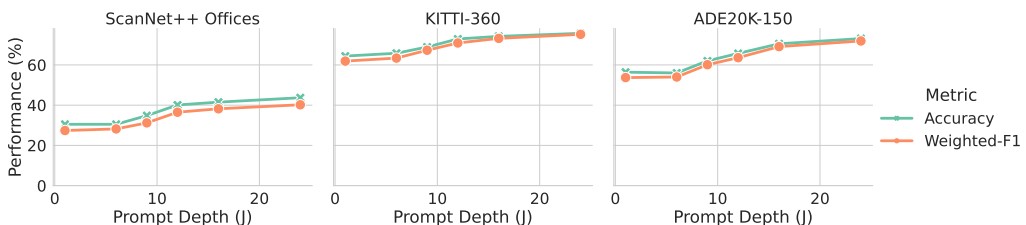

**Figure 13: Prompt Depth.** We compare different prompt depth values on the ScanNet++ Offices, KITTI-360, and ADE20K-150 validation splits. Our further analysis reveals that the more layers we add prompts, the better OpenDAS performs.

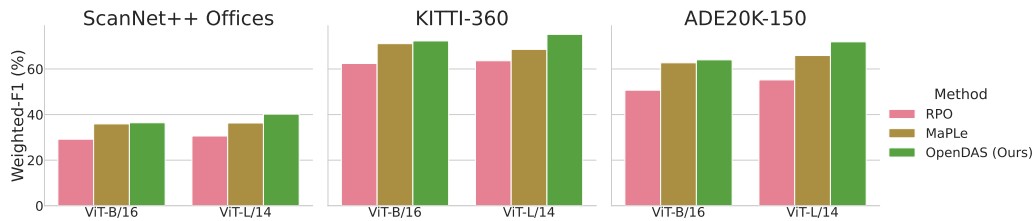

**Figure 14: ViT Backbone.** We compare different ViT backbones on ScanNet++ Offices, KITTI-360, and ADE20K-150 validation splits to observe their impact on performance. ViT-L/14 is significantly larger with 307M parameters compared to ViT-B/16 with 86M parameters. Results show that using a larger backbone boosts the performance in all methods over all datasets.

**Prompt Depth.** We compare different prompt depth, *i.e.*, the number of layers we add prompts, on the ScanNet++ Offices, KITTI-360, and ADE20K-150 validation splits in Fig. 13. Our further analysis reveals that the more layers we add prompts, the better OpenDAS performs over all datasets.

**ViT Backbone.** We present a comparison of the visual backbone and its impact on the accuracy in Fig. 14. We observe that with all multi-modal prompt learning methods, the performance increases with the larger backbone. Hence, we set ViT-L/14 for all our experiments contrary to the prior prompt learning methods' standard settings.

**Number of Learnable Prompts (K).** This determines the number of prompts injected at each layer, or in other words, the 'width' of the prompts. In Tab. 8, we observe that we can gain additional improvement by adding more learnable prompts to the visual encoder for the ScanNet++ Offices and ADE20K-150 validation set. We see that this parameter needs to be tuned for each dataset. However, we choose (K 🎨, K 🔲) = (8,4) as the standard setting for simplicity and to keep the number of parameters minimal.

| K 🍯 | K 📘 | # Params | ScanNet++ Acc | ScanNet++ W-F1 | KITTI-360 Acc | KITTI-360 W-F1 | ADE20K-150 Acc | ADE20K-150 W-F1 |
|---|---|---|---|---|---|---|---|---|
| 8 | 4 | ∼ 135K | 40.1 | 36.5 | **72.9** | **70.9** | 65.7 | 63.6 |
| 8 | 8 | ∼ 172K | 39.6 | 35.5 | 72.0 | 70.4 | 59.3 | 57.2 |
| 8 | 12 | ∼ 209K | 39.7 | 37.0 | 72.3 | 70.9 | 57.5 | 55.2 |
| 12 | 4 | ∼ 184K | **40.3** | **37.4** | 72.4 | 70.7 | 70.1 | 68.5 |
| 12 | 8 | ∼ 221K | 39.6 | 36.7 | 72.7 | 71.5 | **70.9** | **69.4** |
| 12 | 12 | ∼ 258K | 39.6 | 37.2 | 72.3 | 71.1 | 70.4 | 68.4 |

Table 8: **Number of Learnable Prompts (K).** We compare OpenDAS with different numbers of learnable prompts on the Scannet++ Office subset, KITTI-360, and ADE20K-150 (when prompt depth $J = 12$). We denote the prompt length for the visual encoder as K 🍯 and for the textual encoder as K 📘.

| Model | Specialist | ADE20K-150 mIoU | fwIoU | mAcc | pAcc |
|---|---|---|---|---|---|
| OVSeg (Liang et al., 2023) | | 29.8 | 57.8 | 48.1 | 69.3 |
| FC-CLIP (Yu et al., 2024) | | 34.3 | 59.9 | 54.2 | 70.9 |
| MAFT+ (Jiao et al., 2024) | | 36.1 | 61.7 | 55.5 | 73.1 |
| OVSeg + OpenDAS (Ours) | ✓ | 35.8 | 64.3 | 51.7 | 76.2 |
| FC-CLIP + OpenDAS (Ours) | ✓ | 37.3 | 64.7 | 57.1 | 75.9 |
| MAFT+ + OpenDAS (Ours) | ✓ | **38.0** | **66.2** | **57.5** | **76.9** |

Table 9: **Performance Comparison with Predicted Masks.** We integrate OpenDAS to prior generalist OVS models, OVSeg (Liang et al., 2023), FC-CLIP (Yu et al., 2024), and MAFT+ (Jiao et al., 2024), for 2D semantic segmentation on ADE20K-150. We observe consistent improvement with our domain-specific mask and text embeddings.

# G  T-SNE VISUALIZATIONS FOR QUERY EMBEDDINGS

In Fig. 15, we present two t-SNE visualizations (van der Maaten & Hinton, 2008) demonstrating the dimensionality reduction of text embeddings derived from CLIP (Radford et al., 2021) and our method OpenDAS. In the first visualization, each point corresponds to a specific text input, with proximity reflecting the model's interpretation of semantic similarity. For example, the close placements of "door" and "door frame", "carpet" and "floor" demonstrates the strong semantic relationship. However, this embedding space has the drawback of label overlap, which could obscure some labels and result in wrong classification during inference.

In Fig. 15 (*bottom*), we see that OpenDAS addresses the issue of entangled representations, which embeds the labels like "carpet" and "floor" further from each other while maintaining their connection in the latent space. OpenDAS enhances the clarity and distinction of the labels, allowing for precise recognition of objects. Also, it learns the domain-specific meaning of polysemous words like "monitor", embedding it closer to "webcam". This shows that OpenDAS successfully learns to discern similar items in the target domain while preserving their relations in the embedding space.

# H  PERFORMANCE COMPARISON WITH PREDICTED MASKS

In Tab. 9, we further compare our approach against existing 2D OVS models, OVSeg (Liang et al., 2023), FC-CLIP (Yu et al., 2024), MAFT+ (Jiao et al., 2024), on ADE20K-150 as it is one of the commonly used datasets for 2D OVS task. As our method is agnostic to the mask proposals, it can complement prior generalist models for domain-specific segment classification. Hence, we can integrate it into prior existing OVS models for better segment classification. We observe that our model demonstrates consistent improvement on the segment classification over prior models.

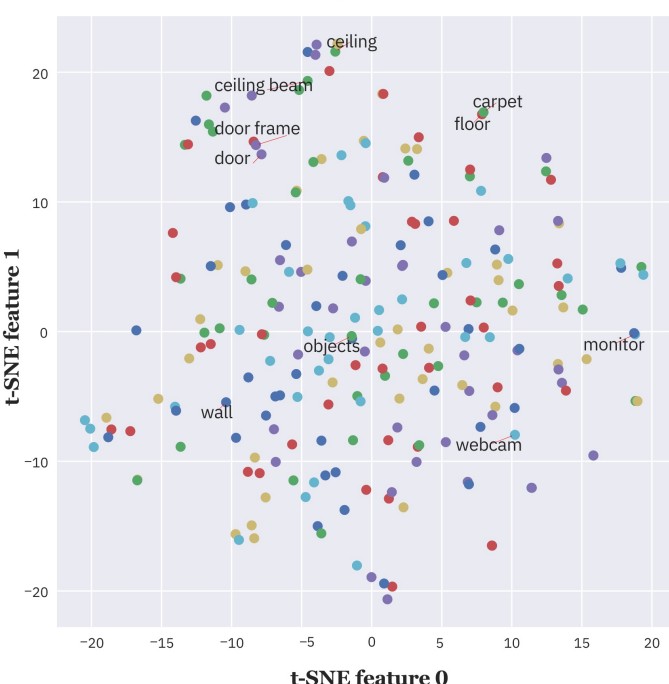

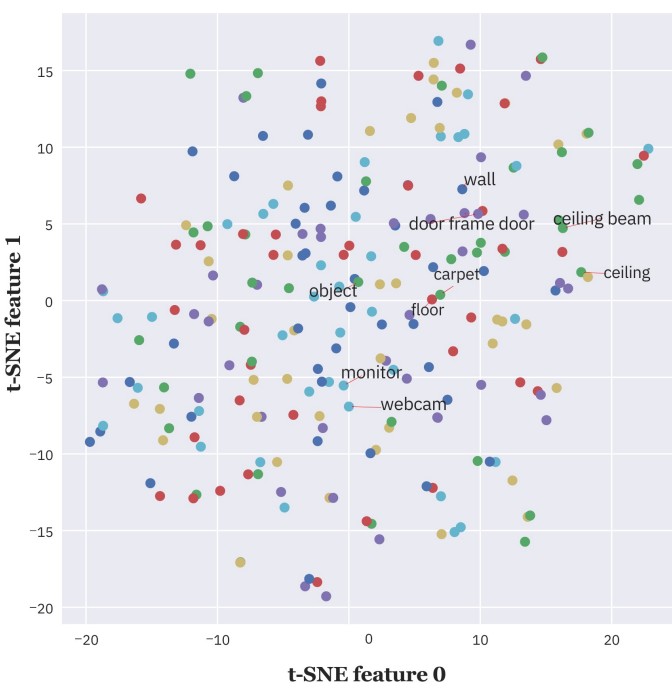

**Figure 15: Comparative t-SNE visualizations.** We compare the text embeddings generated with CLIP (Radford et al., 2021) and OpenDAS (ours). We observe that closely related classes like "door" - "door frame" and "carpet" - "floor" in CLIP's embedding space become more distinct, preventing the model from confusing those classes.

