# OpenReview forum: "OpenDAS: Open-Vocabulary Domain Adaptation for Segmentation"
_ICLR.cc/2025/Conference — Submitted to ICLR 2025_

### Official Review · Reviewer_5sNt · 2024-11-02

**Soundness:** 3
**Presentation:** 2
**Contribution:** 2
**Rating:** 5
**Confidence:** 3

**Summary:**

The paper proposes a new task called open-vocabulary domain adaptation for segmentation. The goal of this task is to improve the recognition performance of vision-language models on a specific domain, while preserving the open-vocabulary recognition capabilities. The proposed method relies on paramater efficient prompt tuning with a combination of cross-entropy and triplet loss. The success of adaptation is measured as classification performance on masked images containing a single segment corresponding to either base or novel semantic class. The experiments are conducted on 2D and 3D segmentation on three different datasets.

**Strengths:**

The proposed method outperforms other adaptation methods on segment classification on both base and novel classes on different benchmarks.
The experiments reveal that the proposed method can also improve the segmentation performance when combined with two existing open-vocabulary segmentation models.

The proposed adaptation approach is simple, sound and effective.

**Weaknesses:**

In my opinion, task name is missleading. The adaptation procedure actually optimizes segment classification (images containing individual segments), while it assumes there is a mask proposal generator available. More precisely, it assumes perfect mask generator both during training and evaluation. The term "open-vocabulary segmentation" implies both localization (mask proposal generation) and classification. Because of that, I was confused while reading the manuscript all the way until section 5 and the description of evaluation metrics. I think that this difference must be clearly stated and thoroughly described earlier in the manuscript.

The manuscript lacks a proper comparison with the related work from open-vocabulary segmentation. Table 3 shows that the adapted model can be used to improve performance of the open-vocabulary segmentation model OVSeg. However, the table shows results only for the ADE20k validation set which contains only in-vocabulary (base) classes, while it is usual to evaluate the performance on datasets with different vocabularies. Furthermore, OVSeg significantly lacks in performance for the current SOTA models (e.g. CatSeg (Cho et al.,2023) or FC-CLIP (Yu et al., 2024)).

The proposed approach is computationally inefficient. For a single full image segmentation it requires multiple forward passes through CLIP backbone in order to classify all the mask proposals. This causes a lot of overhead due to processing potentially overlapping (or empty) image regions. On the other hand, some open-vocabulary segmentation methods avoid this by extracting features with CLIP backbone once, and then pooling these features for each mask (FC-CLIP, OpenSeg, ODISE). The proposed approach might also hurt the classification performance by removing the context (background) from each segment image. It is not clear what are the real performance relations between the two mentioned approaches. Hence, the proper comparison with related methods from open-vocabulary segmentation is necessary.

It is unclear if other open-vocabulary segmentation methods which do not rely on image segment classification (e.g. FC-CLIP) could benefit from the proposed adaptation. This needs further investigation.

**Questions:**

/

---

> ### Author Response · Authors · 2024-11-28
> **Official Comment -- Part 1**
>
> We thank the reviewer for acknowledging the simplicity and effectiveness of our method. We take every comment seriously and hope our response can address the reviewer’s concerns. If there are any remaining questions, we are more than happy to address them.
>
> > **Q1. In my opinion, task name is missleading. The adaptation procedure actually optimizes segment classification (images containing individual segments), while it assumes there is a mask proposal generator available. More precisely, it assumes perfect mask generator both during training and evaluation. The term "open-vocabulary segmentation" implies both localization (mask proposal generation) and classification. Because of that, I was confused while reading the manuscript all the way until section 5 and the description of evaluation metrics. I think that this difference must be clearly stated and thoroughly described earlier in the manuscript.**
>
> The task name is chosen to fit the definition of the task that we introduce in Section 3, independently of the studied method and integrated models which can be considered a first proposal of how to approach the task. Prior works already decoupled the task of OVS into segment proposal and segment classification [1, 2]. Our preliminary analysis shows that the limitation of such decoupled OVS models for the setting of domain adaptation is the segment classification and not mask proposal. Therefore, we focus our proposed method on adapting the segment classification, while showing that integrating the adapted VLM with existing mask-proposal models can indeed improve the more general task of Domain Adaptation for Open-Vocabulary Segmentation. We update the manuscript to better emphasize this (**L99-L101**).
>
> [1] Liang et al., Open-vocabulary semantic segmentation with mask-adapted clip, CVPR'23.
>
> [2] Takmaz et al., OpenMask3D: Open-Vocabulary 3D Instance Segmentation, NeurIPS'23.
>
> > **Q2. The manuscript lacks a proper comparison with the related work from open-vocabulary segmentation. Table 3 shows that the adapted model can be used to improve performance of the open-vocabulary segmentation model OVSeg. However, the table shows results only for the ADE20k validation set which contains only in-vocabulary (base) classes, while it is usual to evaluate the performance on datasets with different vocabularies.**
>
> Thanks for your observation. Table 3 shows that our method can be used with predicted masks instead of just ground truth ones and can be integrated to existing OVS models for better segment classification. To further strengthen this analysis, we have now added results with FC-CLIP, a more recent model, showing that OpenDAS also effectively integrates with and improves this framework. You can also find the updated table with results on ADE20K-150 below.
>
> | Method         | mIoU (%) | fwIoU (%) |
> |----------------|----------|-----------|
> | OVSeg         | 29.8     | 57.8      |
> | + OpenDAS     | **35.8** (+6.0) | **64.3** (+6.5) |
> | **FC-CLIP**   | 34.3     | 59.9      |
> | + OpenDAS     | **37.3** (+3.0) | **64.7** (+4.8) |
>
> Additionally, Table 3 already shows evaluation with predicted masks on **novel queries** in ScanNet++ Offices, where it improves upon the OpenMask3D framework (see below). However, we agree that extending this evaluation to 2D segmentation for novel queries with predicted masks would further strengthen the study.  We are currently preparing these additional results and will update them shortly here in the discussion.
>
> | Method         | AP   | AP₅₀  | AP₂₅  |
> |----------------|------|--------|-------|
> | OpenMask3D    | 8.1  | 11.5   | 14.1  |
> | + OpenDAS     | **12.2** (+4.1) | **18.0** (+6.5) | **24.0** (+9.9) |
>
> > **Q3. Furthermore, OVSeg significantly lacks in performance for the current SOTA models (e.g. CatSeg (Cho et al.,2023) or FC-CLIP (Yu et al., 2024)).**
>
> Based on the suggestion, we look further into integrating our model into newer OVS methods, FC-CLIP [3] and MAFT+ [4]. We observe that OpenDAS shows consistent improvement over existing methods on ADE20K-150, even when it predicts from masked images. We also added these additonal results to the manuscript (see **Appendix H**).
>
> | Method                  | mIoU (%) | fwIoU (%) | mAcc (%) | pAcc (%) |
> |-------------------------|----------|-----------|----------|----------|
> | FC-CLIP                 | 34.3     | 59.9      | 54.2     | 70.9     |
> | FC-CLIP + OpenDAS (Ours)| **37.3**     | **64.7**      | **57.1**     | 75.9     |
> | MAFT+           | 36.1     | 61.7      | 55.5     | 73.1     |
> | MAFT+ + OpenDAS (Ours)| **38.0**     | **66.2**      | **57.5**     | **76.9**     |
>
> [3] Yu et al., Convolutions Die Hard: Open-Vocabulary Segmentation with Single Frozen Convolutional CLIP, NeurIPS'23.
>
> [4] Jiao et al., Collaborative Vision-Text Representation Optimizing for Open-Vocabulary Segmentation, ECCV'24.

---

> > ### Author Response · Authors · 2024-11-28
> > **Official Comment -- Part 2**
> >
> > > **Q4. The proposed approach is computationally inefficient. For a single full image segmentation it requires multiple forward passes through CLIP backbone in order to classify all the mask proposals. This causes a lot of overhead due to processing potentially overlapping (or empty) image regions. On the other hand, some open-vocabulary segmentation methods avoid this by extracting features with CLIP backbone once, and then pooling these features for each mask (FC-CLIP, OpenSeg, ODISE). The proposed approach might also hurt the classification performance by removing the context (background) from each segment image. It is not clear what are the real performance relations between the two mentioned approaches. Hence, the proper comparison with related methods from open-vocabulary segmentation is necessary.**
> >
> > We agree with the reviewer about computational inefficiency compared to coupled methods. Our work builds upon existing frameworks such as OVSeg [1] and OpenMask3D [2], which have already adopted a decoupled approach for mask proposal generation and matching with language queries. As a result, the computational inefficiencies mentioned are inherent to these prior works.
> >
> > However, by integrating our approach with coupled models like FC-CLIP [3] and MAFT+ [4], we demonstrate that these models can still benefit significantly from improved semantic understanding of the masks themselves. This improvement is achieved even without relying on the  background for more context.

---

> ### Author Response · Authors · 2024-12-02
> **Official Comment -- Part 3**
>
> In response to Q2, we share additional results for open-vocabulary 2D semantic segmentation on **ScanNet++ Offices** using OVSeg predicted masks. These results allow us to demonstrate the strength of our method on novel classes, even when the model predicts from the predicted masks. Further results with FC-CLIP and MAFT+ mask proposals will be included in the camera-ready version.
>
> | Method         | mIoU (%) | fwIoU (%) |
> |----------------|----------|-----------|
> | OVSeg          | 7.2      | 36.1      |
> | + OpenDAS      | **10.1** (+2.9) | **50.9** (+14.8) |

---

> > ### Comment · Reviewer_5sNt · 2024-12-02
> >
> > I would like to thank the authors for their effort during the rebuttal. The presented results provide more convincing proofs of the effectiveness of the proposed method. Nevertheless, I am still not convinced that the proposed task substantially differs from the current OVS setup. The same issue was noticed by the reviewers iNo1 and XEDS. Thus, I improved my score, but It still remains below the acceptance threshold.

---

> ### Author Response · Authors · 2024-12-03
> **Official Comment -- Part 4**
>
> We would like to thank the reviewer for reconsidering their evaluation based on our rebuttal. We recognize the importance of further clarifying the distinction between open-vocabulary domain adaptation (OVDA) and open-vocabulary segmentation (OVS), and we provide additional clarification below.
>
> OVS is designed to be a generalist model and handle domain gaps by definition. In a prior work [1], cross-dataset setting for open-vocabulary segmentation is defined as follows:
>
> > **Cross-Dataset Setting.** In this setting, the model is trained on one dataset
> and evaluated on another dataset without fine-tuning. This is a more challenging
> setting than the zero-shot setting, where the model not only deals with the unseen
> classes, but also has to address the **domain gap** among different datasets.
>
> However, OVDA for segmentation focuses on adapting OVS models to **specialized domains** while keeping their ability to respond to **unseen queries** and aims to infuse domain-specific priors to the existing models (as defined in Sec. 3). These domains could be indoor scenes, like offices and homes, or outdoor scenes like urban driving.
>
> We hope this explanation further clarifies the distinction between OVDA and OVS. We understand that this should be better reflected in Sec. 3, where we define OVDA. This will be included in the final version of the paper.
>
> [1] Xu et al., A Simple Baseline for Open-Vocabulary Semantic Segmentation with Pre-trained Vision-Language Model, ECCV'22.

---

### Official Review · Reviewer_iNo1 · 2024-11-03

**Soundness:** 3
**Presentation:** 3
**Contribution:** 2
**Rating:** 5
**Confidence:** 5

**Summary:**

This submission tends to address the task of the open vocabulary domain adaptation to infuse domain specific knowledge into vision language models. Extensive experiments are conducted.

**Strengths:**

The motivation in this paper is convincing, addressing the issue of the lack of pixel-aligned training for pre-trained VLMs.

**Weaknesses:**

1) The "lack of pixel-aligned training masks for VLMs (which are trained on image-caption pairs)" has been extensively studied in previous work [1*, 2*]. The authors should analyze and compare with [1*, 2*], even though they use different optimization strategies

2) This work introduces the concept of Domain Adaptation in OVS. The authors need to clarify the distinction between Domain Adaptation OVS and standard OVS. OVS itself can segment any text and has a general concept of data domains. In experimental settings, the authors use cross-dataset evaluation, which does not differ from the standard OVS settings.

3) In Table 3, the authors only compare with the CVPR23 work, which is insufficient. The latest OVS works [3*, 4*, 5*] should be included for comparison.

4) I have questions about the process of using GPT to generate Negative Queries (specifically the example from "ceiling" to "ceiling fan" in L303). Does this require image input? If only text input is provided, I believe GPT would return synonyms instead.

5) Minor Weaknesses: The writing contains redundancy and needs optimization. e.g., L227-231 does not describe the authors' proposed method; I suggest placing this part in Sec. 4.1.


[1*] Learning mask-aware clip representations for zero-shot segmentation, NeurIPS 23
[2*] Collaborative Vision-Text Representation Optimizing for Open-Vocabulary Segmentation, ECCV 24
[3*] SED: A Simple Encoder-Decoder for Open-Vocabulary Semantic Segmentation, CVPR 24.
[4*] Convolutions Die Hard: Open-Vocabulary Segmentation with Single Frozen Convolutional CLIP, NeurIPS 23.
[5*] Open-Vocabulary Panoptic Segmentation with Text-to-Image Diffusion Models, CVPR 23.

**Questions:**

Please refer to the weaknesses.

---

> ### Author Response · Authors · 2024-11-28
> **Official Comment -- Part 1**
>
> We would like to sincerely thank the reviewer for the detailed and constructive comments on our work. We take every comment seriously and hope our response can address the reviewer’s concerns. If there are any remaining questions, we are more than happy to address them.
>
> > **Q1. The "lack of pixel-aligned training masks for VLMs (which are trained on image-caption pairs)" has been extensively studied in previous work [1, 2]. The authors should analyze and compare with [1, 2], even though they use different optimization strategies**
>
> We appreciate the reviewer for highlighting this point. Our model is designed as a specialist model that can complement generalist models effectively. For instance, the concurrent work MAFT+ [2] also focuses on adapting CLIP for open-vocabulary segmentation tasks. Notably, our model can integrate with such approaches, providing additional improvements for domain-specific queries. Below, we present a direct comparison of our model on ADE20K-150 with the generalist models referenced in [1, 2].
>
> | Method                | mIoU (%) | fwIoU (%) | mAcc (%) | pAcc (%) |
> |-----------------------|----------|-----------|----------|----------|
> | ODISE [1]            | 29.9     | -         | -        | -        |
> | MAFT+ [2]            | 36.1     | 61.7      | 55.5     | 73.1     |
> | MAFT+ + OpenDAS (Ours)| **38.0**     | **66.2**      | **57.5**     | **76.9**     |
>
>
> > **Q2. This work introduces the concept of Domain Adaptation in OVS. The authors need to clarify the distinction between Domain Adaptation OVS and standard OVS. OVS itself can segment any text and has a general concept of data domains. In experimental settings, the authors use cross-dataset evaluation, which does not differ from the standard OVS settings.**
>
> Thanks for your question. Domain adaptation for OVS is conceptually similar to how domain adaptation for closed-set semantic segmentation relates to closed-set segmentation. It should be considered a **sub-problem** of OVS, because as in any other learning task dependent on the scenario, there is a practical interest to trade in overall generalization for higher performance on a better defined domain.
>
> For instance, in scenarios such as deployment of a robot or XR headset in a factory, the environment domain is well-defined and the model can adapt to its environment while maintaining the ability to respond to verbal commands from users.
>
> Our work addresses these practical needs by proposing the novel task of open-vocabulary domain adaptation. Through our analysis, we demonstrate that standard OVS methods have inherent generalization limits (as expected), and their performance can be improved by performing domain adaptation. While neither domain adaptation nor open-vocabulary segmentation alone fully addresses these issues, open-vocabulary domain adaptation bridges this gap by combining the strengths of both paradigms.
>
> > **Q3. In Table 3, the authors only compare with the CVPR23 work, which is insufficient. The latest OVS works [3, 4, 5] should be included for comparison.**
>
> We thank the reviewer for their suggestion. For further analysis, we integrate our domain-specific model into FC-CLIP [4] and a concurrent work MAFT+ mentioned above in [2]. We utilize OpenDAS to match segments upon their predicted masks. We observe that even without ground truth masks, our model can build upon generalist models’ prediction and achieve consistent improvement on ADE20K-150.
>
> | Method                  | mIoU (%) | fwIoU (%) | mAcc (%) | pAcc (%) |
> |-------------------------|----------|-----------|----------|----------|
> | FC-CLIP  [4]              | 34.3     | 59.9      | 54.2     | 70.9     |
> | FC-CLIP + OpenDAS (Ours)| **37.3**     | **64.7**      | **57.1**     | **75.9**     |
> | MAFT+  [2]          | 36.1     | 61.7      | 55.5     | 73.1     |
> | MAFT+ + OpenDAS (Ours)| **38.0**     | **66.2**      | **57.5**     | **76.9**     |

---

> > ### Author Response · Authors · 2024-11-28
> > **Official Comment -- Part 2**
> >
> > > **Q4. I have questions about the process of using GPT to generate Negative Queries (specifically the example from "ceiling" to "ceiling fan" in L303). Does this require image input? If only text input is provided, I believe GPT would return synonyms instead.**
> >
> > The prompt to GPT-4 is **text-only**. To clarify, we list the exact prompt to GPT-4 as well as examples of the generated negative queries in **Appendix C**:
> >
> > * "wall" -> ["room divider", "partition", "divider screen", "privacy screen", "decorative panel"]
> > * "ceiling" -> ["chandelier", "pendant light", "skylight", "light fixture", "ceiling fan"]
> > * "folder organizer" -> ["bedside table", "end table", "chest of drawers",  "bar stool", "storage ottoman"]
> >
> > In some of these cases (e.g. “wall” and “room divider”), the negative query has a similar meaning, even though they are not synonyms and they can not be used interchangeably. This highlights GPT-4’s ability to generate words that are contextually related but semantically distinct.
> >
> > Often, we observe that the generated negative samples are **more specific variations** than the query. However, the majority of negative examples are rather words that appear in the same context but mean something different (e.g. negative examples for ceiling are things that are fixed to a ceiling or even replace a ceiling, such as a skylight). These negative examples help in training models to differentiate between related but distinct concepts, making the process effective for our task.
> >
> > > **Q5. Minor Weaknesses: The writing contains redundancy and needs optimization. e.g., L227-231 does not describe the authors' proposed method; I suggest placing this part in Sec. 4.1.**
> >
> > We thank the author for their suggestion. We updated the manuscript based on the suggestion to reduce redundancy in L227-231. We are open to further suggestions.

---

### Official Review · Reviewer_4Ucz · 2024-11-04

**Soundness:** 3
**Presentation:** 3
**Contribution:** 2
**Rating:** 6
**Confidence:** 5

**Summary:**

This paper introduces OpenDAS, a novel approach to open-vocabulary domain adaptation (OVDA) for 2D and 3D segmentation. Traditional segmentation models are typically limited to closed-set vocabularies, lacking flexibility for novel classes or domain-specific knowledge. OpenDAS addresses this by adapting Vision-Language Models (VLMs), such as CLIP, to domain-specific segmentation tasks without sacrificing their open-vocabulary capabilities. The authors propose a method combining prompt tuning with a triplet-loss-based training strategy, incorporating auxiliary negative queries to boost generalization to novel classes within a target domain. OpenDAS integrates seamlessly with existing segmentation models, such as OVSeg for 2D and OpenMask3D for 3D segmentation, and experimental results show significant improvements over baseline methods on various benchmarks.

**Strengths:**

* It defines a new task that adaptes VLMs to domain-specific segmentation tasks while preserving open-vocabulary capabilities.

* Extensive experiments demonstrate adaptability for both 2D and 3D segmentation tasks, improving performance on existing pipelines without extensive modifications.

**Weaknesses:**

* The proposed framework is engineering, because the triple loss, visual and textual prompt tuning are common ways in open-vocabulary and domain adaptation communities.

* More recent vision-language models should be compared, like SigLIP, llama, etc.

* Several ablation studies are needed, like the discussion about the margin $\mu$.

* The writing and structure should be further improved, e.g., the pipeline fig is a bit ambiguous.

**Questions:**

Please see weaknesses.

---

> ### Author Response · Authors · 2024-11-28
>
> We would like to sincerely thank the reviewer for the detailed and constructive comments on our work. We take every comment seriously and hope our response can address the reviewer’s concerns. If there are any remaining questions, we are more than happy to address them.
>
> > **Q1: The proposed framework is engineering, because the triple loss, visual and textual prompt tuning are common ways in open-vocabulary and domain adaptation communities.**
>
> We respectfully disagree with the notion that our framework is solely an engineering contribution. While our approach incorporates established techniques like triplet loss and prompt tuning, the key contribution lies in their integration into a unified framework specifically tailored for the **novel task of open-vocabulary domain adaptation**.
>
> This task addresses critical real-world challenges that are not adequately addressed by existing paradigms, whether in open-vocabulary segmentation or domain adaptation, do not adequately resolve. By bridging these contexts, our framework introduces a new perspective that goes beyond the scope of traditional methods.
>
> Furthermore, we conduct extensive comparisons with prior visual and textual prompt-tuning methods (see Table 1 and 2), showing a clear and significant advantage of the proposed framework in addressing this problem.
>
> > **Q2: More recent vision-language models should be compared, like SigLIP, llama, etc.**
>
> Thank you for your suggestion. Our paper focuses on the general setting of open-vocabulary segmentation methods. As CLIP is the standard VLM used by most SOTA OVS models [1,2,3,4,5], we prioritize adapting CLIP to ensure easy integration and comparison to existing  models in this domain.
>
> We acknowledge the value of exploring more recent vision-language models like SigLIP and LLaMA, and we consider this an exciting direction for future research to further enhance our framework.
>
> [1] Liang et al., Open-vocabulary semantic segmentation with mask-adapted clip, CVPR'23.
> [2] Xu et al., Side Adapter Network for Open-Vocabulary Semantic Segmentation, CVPR'23.
> [3] Cho et al., CAT-Seg: Cost Aggregation for Open-Vocabulary Semantic Segmentation, CVPR'24.
> [4] Yu et al., Convolutions Die Hard: Open-Vocabulary Segmentation with Single Frozen Convolutional CLIP, NeurIPS'23.
> [5] Jiao et al., Collaborative Vision-Text Representation Optimizing for Open-Vocabulary Segmentation, ECCV'24.
>
> > **Q3: Several ablation studies are needed, like the discussion about the margin μ**
>
> We thank the reviewer for their suggestion to look further into margin $\mu$. Our initial analysis had shown that $\mu=1.5$ was better than $\mu=1.0$ (default value in Pytorch) for domain adaptation. Now, we further look into the effect of  $\mu$ and we add the results to the **Appendix F**. You can find the Weighted-F1 scores below. We observe that higher margin is only better when we test the model with the base classes. When there are novel classes, the model performance drops heavily after a certain threshold, which is in this case > 1.5.
>
> | μ   | ADE20K-150       | ADE20K → SN++ Offices | ADE20K → KITTI-360  |
> |-----|------------------|-----------------------|---------------------|
> | 1.0 | 71.3     | 20.1      | 36.6     |
> | **1.5** | 71.9     | **23.0**      | **47.1**     |
> | 2.0 | **72.5**     | 12.9      | 35.8    |
>
> We are open to further suggestions.
>
> > **Q4: The writing and structure should be further improved, e.g., the pipeline fig is a bit ambiguous.**
>
> We are always eager to further improve our writing and make sure the paper is well-readable to everyone. While we focused our efforts during the discussion phase on the additional results, we would be very thankful for more input on what is currently ambiguous about Figure 2.

---

> > ### Comment · Reviewer_4Ucz · 2024-12-03
> >
> > I agree with the authors that a new task has been proposed, which I also mentioned in Strengths. However, the method is mainly a combination of previous approaches. Thus, I still think this paper is borderline for ICLR. As most of my concerns were solved, I improve my score to borderline accept.

---

### Official Review · Reviewer_XEDS · 2024-11-07

**Soundness:** 2
**Presentation:** 3
**Contribution:** 2
**Rating:** 3
**Confidence:** 4

**Summary:**

This paper introduces the task of domain adaptation for language-vision models in open-vocabulary 2D and 3D segmentation. In this framework, language-vision models are fine-tuned in a weakly supervised manner to enhance performance on domain-specific segmentation tasks.

The authors propose a method specifically tailored for the CLIP architecture. A pretrained CLIP model is extended by adding trainable prompts to the inputs of selected layers in both the textual and visual encoders, with the number of layers modified as a hyperparameter. The optimization process occurs in two phases. In the first phase, only the visual prompts are trained: an image passes through the visual encoder, while a set of domain-specific labels—comprising both positive and ChatGPT-generated negative queries—passes through the textual encoder. A cross-entropy loss is used to bring the image embedding closer to the correct textual embedding. In the second phase, only the textual prompts are trained, using a combined cross-entropy and triplet loss. For the triplet loss, a hard negative sample mining strategy is employed to refine results.

The authors conduct experiments on both 2D and 3D segmentation tasks, along with ablation studies to analyze the impact of different components in the proposed approach.

**Strengths:**

S1) Paper is clearly written

S2) Experimental results demonstrate improvements over baseline CLIP performance when integrated with the OVSeg model. Additionally, the proposed prompt-learning method for specializing vision-language models (VLMs) outperforms alternative approaches

**Weaknesses:**

W1 The paper diverges from the established definition and test setup for open-vocabulary segmentation. While previous studies assume that a segmentation model fine-tuned on one domain should retain its generalizability across other domains, this work fine-tunes on a more general domain and evaluates on a narrower domain. Consequently, the test setup is not directly comparable to previous open-vocabulary approaches, such as those in (Liang et al. 2023)

W2 The fine-tuning approach assumes access to densely labeled, domain-specific images, which differs from previous domain adaptation frameworks that often use self-supervised or student-teacher setups. Here, the fine-tuning process effectively performs domain-specific classification training.

**Questions:**

1) What are the relations between ADE20K, SN++ Offices and KITTI-360 classes in Table 2?
2) Were the image segments used for "domain adaptation" obtained from densely labelled images?

---

> ### Author Response · Authors · 2024-11-28
>
> We thank the reviewer for acknowledging the clarity of our writing and recognizing the strong experimental results. We take every comment seriously and hope our response can address the reviewer’s concerns. If there are any remaining questions, we are more than happy to address them.
>
> > **Q1: The paper diverges from the established definition and test setup for open-vocabulary segmentation. While previous studies assume that a segmentation model fine-tuned on one domain should retain its generalizability across other domains, this work fine-tunes on a more general domain and evaluates on a narrower domain. Consequently, the test setup is not directly comparable to previous open-vocabulary approaches, such as those in (Liang et al. 2023)**
>
> Thank you for the question. We agree that our setup differs from the established definition and test setup for open-vocabulary segmentation. This difference was intentional and motivated by the limitations observed in existing setups, where generalizability across domains remains a significant challenge (see Fig. 1, Tab. 1 and 2 for no adaptation results).
>
> To address these limitations and better reflect real-world use cases, we introduce a new setup for domain-adaptive open-vocabulary segmentation, where a model adapts to its environment while retaining the ability to respond to any language query from the user. This hybrid scenario lies at the intersection of domain adaptation and open-vocabulary segmentation, addressing gaps that neither field adequately covers.
>
> We understand that this new setup is not directly comparable to prior open-vocabulary approaches. However, we argue that **exploring relevant real-world problems, especially ones that have not been previously studied, represents an important scientific contribution rather than a weakness**.
>
> Our test setup was specifically designed to address these real-world user needs, which is also why we created our custom split from ScanNet++. The official splits do not sufficiently capture the scenarios we aim to address.
>
> > **Q2: The fine-tuning approach assumes access to densely labeled, domain-specific images, which differs from previous domain adaptation frameworks that often use self-supervised or student-teacher setups. Here, the fine-tuning process effectively performs domain-specific classification training.**
>
> Our approach is motivated by real-world applications where access to densely labeled images in the target domain is often feasible such as deployment of a robot or XR headset in a factory. In these settings the model is required to learn in-domain language queries while answering to novel queries.
>
> Unlike traditional domain adaptation frameworks, which rely on self-supervised or student-teacher setups, our method employs a unique hybrid framework inspired by supervised domain adaptation (SDA) and open-vocabulary segmentation. Specifically, we leverage domain-specific images as in adaptation approaches for foundational models [1], to better align the model with the target domain.
>
> While SDA methods typically utilize small datasets for adaptation [2], we ensure that our dataset remains relatively small compared to the original vision-language model (VLM) training scale. This approach maintains the model's generalization capability while enabling effective domain adaptation. We further clarify our approach in the revised version (see **L107-L108**).
>
> [1] Ha et al., Domain-Aware Fine-Tuning: Enhancing Neural Network Adaptability, AAAI’24
> [2] Wang et al., Deep visual domain adaptation: : A survey, Neurocomputing'18
>
> > **Q3: What are the relations between ADE20K-150, SN++ Offices and KITTI-360 classes in Table 2?**
>
> To test adaptation on both base and novel queries, we use datasets with overlapping but not identical category sets. This approach allows us to evaluate our method across diverse visual domains, while ensuring a fair test of generalization to unseen categories.
>
> * **ScanNet++ Offices Split:** The best test case for the target scenarios described in the introduction is our custom ScanNet++ Offices split. This curated subset of ScanNet++ ensures a significant number of test categories that never appear in the training set, providing a robust evaluation for novel category segmentation.
> * **Cross-Dataset Evaluation:** To further test adaptation across distinct visual domains and facilitate comparisons on established datasets, we include cross-dataset evaluations using ADE20K, KITTI-360, and ScanNet++ Offices.
>
> **ADE20K-150** spans both indoor and urban scenes, creating natural overlaps with KITTI-360 (urban scenes) and ScanNet++ (indoor scenes). Most frequently appearing classes in KITTI-360 and ScanNet++ Offices also overlap with ADE20K-150 categories in different forms:
> * 18 out of 37 classes in KITTI-360 also appear in ADE20K-150.
> * 47 out of 233 test classes in ScanNet++ offices also appear in ADE20K-150.
>
> We update the manuscript to further clarify this (see **L373-L377**).

---

> ### Author Response · Authors · 2024-11-28
> **Official Comment -- Part 2**
>
> > **Q4: Were the image segments used for "domain adaptation" obtained from densely labelled images?**
>
> Yes, this setup aligns closely with how supervised domain adaptation (SDA) is typically studied [1, 2, 3, 4]. Our work is motivated by scenarios where densely labeled target domain images are available, but the dataset is not large enough to cover all possible user queries. Therefore, we adapt large-scale pre-trained VLMs to the target domain using a relatively small amount of data compared to the original training dataset.
>
> In our custom ScanNet++ offices split, designed specifically for our task, we use fewer labeled images for training and evaluate on more diverse scenes, including those with unseen objects. This setup reflects our task definition, where the dataset cannot fully cover all possible language queries.
>
> We also agree with the reviewer that exploring an unsupervised variant of this problem is a promising direction. This is an area we are actively researching as a follow-up to this work.
>
> [1] Wang et al., Deep Visual Domain Adaptation: A Survey, Neurocomputing'18.
> [2] Sun et al., Not All Areas Are Equal: Transfer Learning for Semantic Segmentation via Hierarchical Region Selection CVPR'19.
> [3] Ha et al., Domain-Aware Fine-Tuning: Enhancing Neural Network Adaptability, AAAI'24.
> [4] Motiian et al., Unified Deep Supervised Domain Adaptation and Generalization, ICCV'17.

---

### Author Response · Authors · 2024-12-01
**General Response to Reviewers and AC**

Dear Reviewers and AC,

We sincerely thank the reviewers and AC for the time they spent on our submission. We take all the comments and suggestions very seriously to improve our work and hope our rebuttal can address these comments and suggestions. We have **updated the manuscript** accordingly. All revisions in the updated version are highlighted in **red**. The revisions in the paper and comments can be summarized as follows:

Add more experimental results:
* Add comparisons to more recent OVS methods: FC-CLIP [1], MAFT+ [2] (iNo1-Q1; iNo1-Q3; 5sNt-Q2; 5sNt-Q3; Table 3 and Appendix H).
* Add comparisons with predicted masks on ScanNet++ Offices (5sNt-Q2, in the comments: "Official Comment -- Part 3").
* Add ablation of mu (4Ucz-Q3; Appendix F).

Add more discussion and details:
* Clarify that the proposed method focuses on better segment classification, rather than adapting the mask proposals (5sNt-Q1; L99-L101).
* Clarify that our approach is based on densely labeled images (XEDS-Q2; L107-L108).
* Move L227-231 from Section 4.2 to 4.1 (iNo1-Q5; now L224-L230).
* Clarify the relationship between ADE20K and ScanNet++ Offices, as well as ADE20K and KITTI-360, respectively (XEDS-Q3; L373-L377).

Thanks again for all the effort and time.

Best,

Authors

[1] Yu et al., Convolutions Die Hard: Open-Vocabulary Segmentation with Single Frozen Convolutional CLIP, NeurIPS'23.

[2] Jiao et al., Collaborative Vision-Text Representation Optimizing for Open-Vocabulary Segmentation, ECCV'24.

---

### Author Response · Authors · 2024-12-02
**Kindly Reminder: Response Deadline Approaching**

Dear Reviewers,

We are thankful for the time reviewers committed to review our submission so far. We would like to remind all the reviewers that the deadline to respond is approaching (2nd December, 23:59 AoE). If there are any remaining questions, we would be happy to provide further clarifications.

Best regards,

Authors.

---

### Meta-Review · Area_Chair_sub5 · 2024-12-08

**Metareview:**

This paper introduces a simple yet effective approach to fine-tune open-vocabulary 2D and 3D segmentation models by leveraging negative prompts generated by LLMs.The strengths of this work lie in its clarity, well-structured presentation, and comprehensive experimental validation across multiple architectures and datasets, which demonstrate the universality of the proposed approach.  The additional results provided during the rebuttal further strengthen the empirical evidence, showing that despite the simplicity of the method, it achieves notable performance improvements.

However, a major concern raised by multiple reviewers pertains to the framing of the proposed OVDA task. The approach assumes access to pixel-level annotations in the target domain but does not adequately explain why novel classes remain unlabeled despite the availability of such dense annotations. This assumption limits the practical applicability of the method, as real-world scenarios often involve either weak supervision (e.g., partially annotated data) or unsupervised settings (e.g., no target labels at all). These scenarios may further introduce the challenges that were not discussed in this paper, such as the domain-specific concept definitions can vary significantly (e.g., "road" vs. "sidewalk"). Achieving robustness across such variations would require much more engineering efforts to adapt ChatGPT to all possible cases. Additionally, the method introduces considerable computational overhead, particularly due to its reliance on multiple forward passes through the CLIP to process overlapping image regions. This inefficiency remains unresolved.

Overall, the proposed method is an engineering-oriented solution, but its novelty and comprehensiveness are somewhat limited. Reframing the task to address more practical scenarios would better position the work in the future.

**Additional Comments On Reviewer Discussion:**

During the rebuttal period, several key points were raised including
- the definition and novelty of the newly proposed tasks (XEDS, iNo1, 5sNt): They found the task name or scope misleading, arguing that it lacks clarity in distinguishing itself from standard open-vocabulary segmentation (OVS) and primarily focuses on segment classification rather than full segmentation. Some questioned whether this setup significantly differs from existing OVS paradigms. While some reviewers acknowledged the clarification provided by authors, others remained unconvinced, suggesting that the novelty is not substantial or clearly articulated.
- comparisons about the SOTA.  Reviewers (4Ucz, iNo1, 5sNt) noted insufficient comparisons with state-of-the-art models like FC-CLIP, SigLIP, and MAFT+. They also felt that the evaluation lacked robustness, especially for novel classes and across different datasets. The authors provided additional results comparing their method with newer models and demonstrated improvements in integration with existing OVS frameworks. Reviewers appreciated the additional results and acknowledged the improvements but continued to express concerns about whether the comparisons were sufficient and comprehensive.
- method and implementation. Reviewers (4Ucz, iNo1, 5sNt) felt the methodology was primarily an engineering solution that combined existing techniques like triplet loss and prompt tuning, without significant innovation. Concerns about computational inefficiency and the decoupling of mask proposals and classification were also raised. The authors argued that the integration of techniques within a unified framework is novel and addresses real-world challenges. They acknowledged computational inefficiencies inherent in decoupled approaches but demonstrated consistent improvements when integrated with existing models. While the simplicity and effectiveness of the method were acknowledged, some reviewers remained skeptical about its innovation and practical feasibility due to computational overhead.

In my final decision, I carefully weighed the strengths and weaknesses highlighted during the review process. While the additional experiments provided by the authors enhanced the convincingness of the proposed approach and demonstrated its effectiveness in domain-specific settings, the underlying concerns regarding the unclear task definition and limited methodological novelty remained significant. These concerns, raised consistently by multiple reviewers, suggest that the contribution of the paper may not be as distinct or impactful as required for acceptance.

---

### Decision · Program_Chairs · 2025-01-22

Reject